# Cell cycle gene regulation dynamics revealed by RNA velocity and deep-learning

Andrea Riba[1,2 ✉], Attila Oravecz[1,2], Matej Durik[1], Sara Jiménez[1], Violaine Alunni[1], Marie Cerciat[1], Matthieu Jung[1], Céline Keime [1], William M. Keyes[1] & Nacho Molina [1 ✉]

Despite the fact that the cell cycle is a fundamental process of life, a detailed quantitative understanding of gene regulation dynamics throughout the cell cycle is far from complete. Single-cell RNA-sequencing (scRNA-seq) technology gives access to these dynamics without externally perturbing the cell. Here, by generating scRNA-seq libraries in different cell systems, we observe cycling patterns in the unspliced-spliced RNA space of cell cycle-related genes. Since existing methods to analyze scRNA-seq are not efficient to measure cycling gene dynamics, we propose a deep learning approach (DeepCycle) to fit these patterns and build a high-resolution map of the entire cell cycle transcriptome. Characterizing the cell cycle in embryonic and somatic cells, we identify major waves of transcription during the G1 phase and systematically study the stages of the cell cycle. Our work will facilitate the study of the cell cycle in multiple cellular models and different biological contexts.

[1] Institut de Génétique et de Biologie Moléculaire et Cellulaire (IGBMC); Université de Strasbourg; Centre National de la Recherche Scientifique (CNRS) UMR 7104; Institut National de la Santé et de la Recherche Médicale (INSERM) UMR-S 1258, 1 Rue Laurent Fries, 67404 Illkirch, France. [2] These authors contributed equally: Andrea Riba, Attila Oravecz. ✉email: arriba87@gmail.com; molinan@igbmc.fr

Cells divide by progressing through highly organized phases in which they grow, synthesize a copy of their genetic material, and, finally, undergo mitosis[1]. Alternatively, cells can stop cycling and reversibly transition into quiescence, irreversibly differentiate or become senescent[2]. These processes require tight dynamic regulation of gene expression and, despite immense research during the past decades, a quantitative picture of the gene regulation dynamics across the cell cycle is still incomplete. With the advent of single-cell RNA sequencing (scRNA-seq), scientists can now analyze intrinsically asynchronous populations of cells enabling the simultaneous identification and analysis of cells at different cell cycle stages. Thus, scRNA-seq provides a high-resolution approach to study the cell cycle without external perturbations, such as synchronization by drugs or engineered fluorescent reporters[3,4]. While many attempts to computationally assign cell cycle phases have been performed[5–8], these typically lack generalizability and fail in accurately capturing cell-cycle dynamics. To overcome these problems we propose to use RNA velocity, an approach that characterizes the transcriptional state of individual genes based on their spliced and unspliced RNA signals in single cells[9]. Briefly, RNA velocity is based on how the process of transcription operates. RNA polymerase II starts transcribing a gene, where it first generates a pre-mRNA molecule that contains both exons and introns. Then, the pre-mRNA molecule undergoes splicing to produce the final mRNA molecule without any introns. The estimation of the pre- and mature mRNA levels from unspliced and spliced reads therefore allows quantification of the transcriptional changes happening at the single-gene level. Then, by studying the distribution of unspliced and spliced reads across all cells (unspliced-spliced RNA space), it is possible to estimate if a specific gene is being transcribed or inactivated in any single cell. This can be simultaneously applied to all genes, in all cells of a population. With such analysis, cell-cycle-related genes would be expected to undergo activation and deactivation phases within a single cell cycle, resulting in a circular pattern in the unspliced-spliced RNA space. Thanks to the depth of the scRNA-seq datasets generated in this study, such circular patterns in the unspliced-spliced RNA space can be observed clearly for a subset of genes, and exploited to naturally stratify cells across the cell cycle. The challenge is then to assign, in a reliable and robust manner, a single parameter to each cell that describes its cell-cycle state combining the information in the cycling patterns. To accomplish this, we designed DeepCycle (https://github.com/andreariba/DeepCycle), a deep learning method to ascribe a continuous high-resolution cell cycle trajectory to single cells based on RNA velocity. The approach applies to different cell types and has self-consistency checks to establish whether the analysis worked properly. Deep-Cycle allows us to fit the dynamics of gene activation and inactivation in the unspliced-spliced RNA space with minimal assumptions, and assign cells to cell cycle stages, generating gene expression series.

Different cell types have specific cell cycle dynamics[10,11]. For example, the cell cycle is deeply affected by the degree of stemness, such that pluripotent and neural stem cells have short G1 phases, while committed cells extend their G1 phases and present with longer overall cell cycles[11–13]. Thanks to DeepCycle, we not only recapitulate these findings in mESCs and human fibroblasts, but also extend the analysis to a public dataset of ductal cell progenitors. This allows us to suggest underlying regulatory mechanisms involved, highlighting different genes and transcription factors that are active in the different cellular models across the cell cycle.

Finally, as most of the cells within multicellular organisms are not actively cycling, tight control over cell cycle entry and exit is critical, as seen for example in embryonic development, hematopoiesis, activation of adaptive immune responses, and wound healing[14]. However, in diseases like cancers, cells do not consistently respond to the normal regulatory cues and signals. It is therefore important to understand the processes that determine cell cycle entry, cell cycle progression, and exit to quiescence[14]. Here, we characterize the branching point where human fibroblasts exit from the cell cycle. This confirms previous findings, and uncovers marker genes and transcription factors underlying the process, paving the way to the systematic characterization of the G1-G0 transition in other cellular models.

## Results

**Generation of deep-sequenced single-cell RNA-seq datasets.** To robustly study the cell cycle, we reasoned that the dataset should be enriched for proliferating cells. The majority of public scRNA-seq datasets have been generated to study the overall population of cells in a given condition, and, typically, they contain heterogeneous cell types. Therefore, we compared three distinct populations of proliferating cells. First, we cultured mouse embryonic stem cells (mESCs) in 2i+LIF medium to maintain the ground state of pluripotency by blocking differentiation[15–17], and generated a scRNA-seq library from more than five thousand mESCs (Fig. 1A). Then, we included ductal cell progenitors from pancreas development in mice in our scRNA-seq analysis (henceforth referred to as *ductal cells*) (Fig. 1B)[18]. These cells have been linked to a proliferative cell state by specific marker genes[19]. Finally, to compare the results to a different cell type from another organism we also sequenced 5367 human fibroblasts (Fig. 1C). These fibroblasts separate into two subpopulations, only one of which expressing cell cycle genes (Supplementary Figs. S1, S2, 16 out of the top 25 genes belong to the DAVID Keywords Cell cycle, Benjamini = 2e-15), therefore we first focused on the proliferative subpopulation ($n = 3086$).

The three datasets (mESCs, ductal cells, and fibroblasts) present different sequencing depths: mESCs and fibroblasts have ~30 thousand unique molecular identifiers (UMI) per cell, median values of 31977 and 27319 UMIs, respectively, a depth that is high for the recent standards; while the ductal cells are as low as 8 thousand UMIs per cell, the median value of 8043 (Fig. 1D). Similarly, the median number of genes identified per cell varies from 2840 in the ductal cells to 5161 and 5630 in fibroblasts and mESCs (Fig. 1D). These differences across samples might be partially explained by their respective sequencing depths: total spliced and unspliced reads of 8 M, 210 M, and 220 M in ductal cells, fibroblasts, and mESCs, respectively (Fig. 1D). Overall, they have similar fractions of unspliced reads (Fig. 1E). All the datasets contain genes with circular patterns in the spliced-unspliced read space in accordance with the RNA velocity theory (Fig. 1F). Cycling genes are expected to be characterized by fully circular patterns as they complete both their activation and deactivation phases (Fig. 2A). Overall, these datasets constitute a unique opportunity to study gene regulation throughout the cell cycle in different mouse and human cell types.

**Inference of a cell-cycle transcriptional phase from single-cell RNA-seq data.** The dynamical state of a gene can be inferred by comparing its unspliced and spliced reads[9]. Unspliced reads indirectly measure the nascent transcripts, and the spliced ones the mature messenger RNAs (see Figs. 1F and 2A). The comparison of the two quantities at the single-cell level allows the inference of the transcriptional activation, or deactivation, of a gene. The original RNA velocity framework proposed by La Manno et al.[9] assumed either constant velocity or constant unspliced molecules; to overcome this limitation, Bergen et al.[19] developed an extension of the original model to include

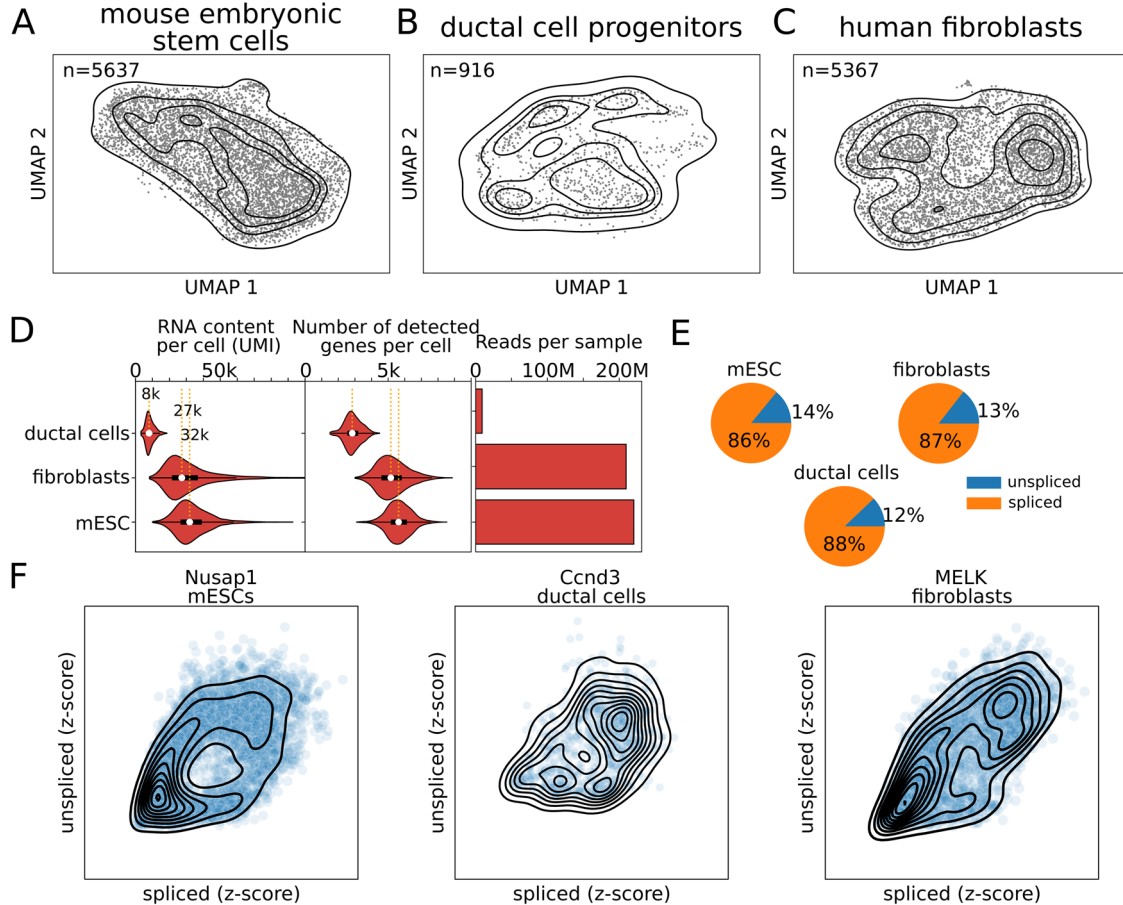

**Fig. 1 Single-cell RNA-sequencing data show the RNA velocity patterns.** UMAP projections for mouse embryonic stem cells (**A**), ductal cell progenitors (**B**), and human fibroblasts (**C**). **D** Distribution of RNA content (UMI), number of genes per cell, and the total number of reads (unspliced + spliced) for each sample. White dots represent the median. Boxes represent 50% of the data and whiskers 99%. **E**. Fractions of spliced and unspliced reads in the three datasets. **F** Examples of the unspliced-spliced patterns for *Nusap1*, *Ccnd3*, and *MELK* in mouse embryonic stem cells, ductal cells, and human fibroblasts, respectively.

intermediate states and more flexible dynamical parameters (scVelo). However, the extended model was unable to fit the actual dynamics for the genes in our datasets, while the inferred latent time did not capture the correct dynamics of the cells (see Supplementary Fig. S3). Therefore, we reasoned that the complexity of gene regulation in the context of the cell cycle cannot be approximated by the current models[9,19] and that a more flexible approach is required. In order to achieve this, we developed a method based on neural networks, taking advantage of their ability to represent a universal function approximator[20].

We expect that genes whose expression is regulated during the cell cycle show a closed path in the unspliced-spliced RNA space consisting of both an active and inactive phase (see Fig. 2A, B). Overall, the cell-cycle progression of a cell can be viewed as a periodic trajectory within the 2 N-dimensional unspliced-spliced space where N is the number of considered genes. This embedded 1-dimensional manifold representing the cell cycle can be characterized by a circular latent variable, the *transcriptional phase* ($\theta$), that maps cells into the particular location of the periodic trajectory. Notice that $\theta$ is a continuous variable representing the continuous cell-cycle progression of cells that has not to be confused with the discrete phases of the cell cycle (G1, S, G2, and M). Then, the estimation of $\theta$ for each cell given the unspliced and spliced reads is an embedded manifold learning problem. To solve this problem, we developed DeepCycle, a deep learning method based on an AutoEncoder (AE) neural network.

AEs are designed to perform non-linear dimensionality reduction by compressing the information contained in the inputs to a lower-dimensional space (latent space) in the encoding phase. The compressed information is then used to reconstruct the original input in the decoding phase. AEs have been used to analyze scRNA-seq data and accomplish different tasks, from clustering to de-noising[21–28]. DeepCycle is constructed as an AE with a single latent variable representing the cell-cycle transcriptional phase $\theta$ that is then transformed with cosine and sine functions in the first layer of the decoder (Fig. 2C and Supplementary Fig. S4).

To train DeepCycle, we used the expression of unspliced and spliced RNAs of the genes in the GOterm:cell_cycle ($n = 532$, see 'Implementation of DeepCycle' in Methods) determining circular paths for cycling genes in the unspliced-spliced space and removing technical noise or biological fluctuations associated with stochastic gene expression (see examples in Fig. 2C, D and Supplementary Fig. S5). Finally, a transcriptional phase is assigned to each cell in the dataset (see 'Implementation of DeepCycle' in Methods) and the dynamics of unspliced and spliced RNA with respect to the transcriptional phase can be further analyzed. It is important to note that the transcriptional phase is a nonlinear monotonic function of time that can be arbitrarily complex, so we cannot directly infer temporal dynamics with it. Importantly, DeepCycle robustly returns very similar transcriptional phases by selecting as input the genes

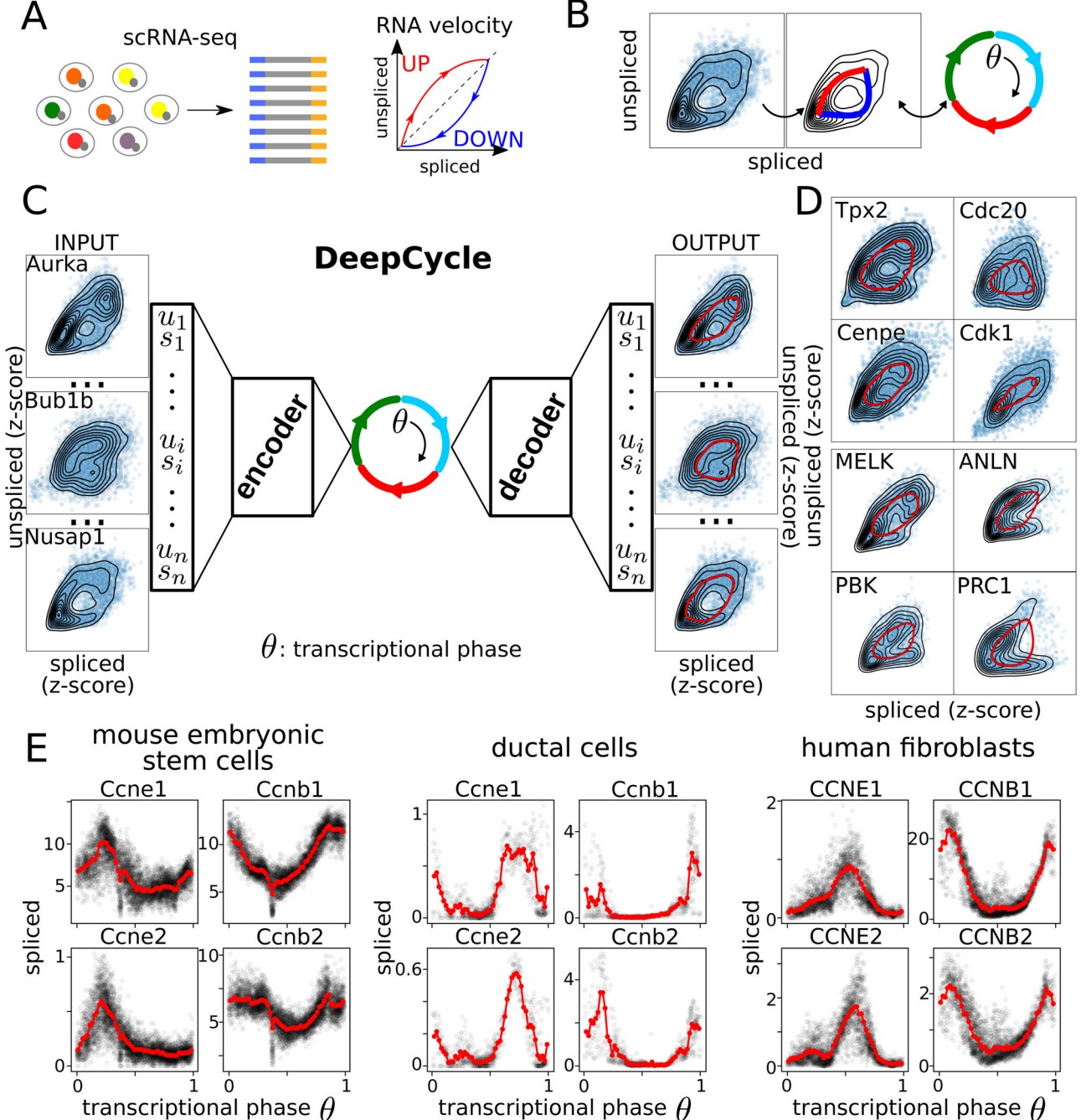

**Fig. 2 Transcriptional phase inference with DeepCycle. A** Single-cell RNA-seq combined with RNA velocity analysis allows the detection of transcriptional changes within a single cell. **B** Fully circular RNA velocity patterns can be mapped to an angle describing the transcriptional state of a gene. By generalizing this to all genes, the angle will describe the actual transcriptional state of a cell. The angle is called the *transcriptional phase*. **C** DeepCycle infers the transcriptional phase of each cell. It takes as input the spliced-unspliced reads ($s_i$, $u_i$) for a set of n genes ($i = 1,...,n$). By fitting the transcriptional phase $\theta$, it can denoise and predict the unspliced-spliced expressions for each transcriptional phase. **D** Examples of cycling genes and fits of the RNA velocity patterns (red lines) in mESCs and human fibroblasts. **E** Cyclin E and B levels along the transcriptional phase in the three datasets. Negative controls for non-cycling genes have been added for comparison in Supplementary Fig. S10.

showing multiple maxima in the unspliced-spliced space (Supplementary Fig. S6 and 'Implementation of DeepCycle' in Methods). The genes presenting multiple maxima ($n = 158$) are listed in Supplementary Data 1 that includes cycling genes not yet considered in the GO term:cell_cycle that could be added as markers of the cell cycle.

Finally, we compared DeepCycle with Cyclum[7], a recent method developed for the analysis of the cell cycle in scRNA-seq data also based on an AE. Strikingly, Cyclum was not able to

place cells consistently in a circular 1D manifold and therefore could not correctly identify the cell-cycle progression of single cells when applied to our datasets (Supplementary Fig. S7). Multiple runs of Cyclum also give inconsistent results that questions its stability (#1 and #2 in Supplementary Fig. S7). As opposed to Cyclum, DeepCycle is based on RNA velocity and trained on both spliced and unspliced RNA levels, which explain the better performance, as shown by the ablation analysis (Supplementary Fig. S8). By removing either the unspliced or

the spliced from the input to DeepCycle, the inferred transcriptional phase becomes inconsistent (see Supplementary Fig. S8 for details).

Another recent method to analyze the cell cycle at the single-cell level is Revelio[8]. This method is based on the inference of a "cylindrical" manifold in the multidimensional gene expression space. It takes as input the list of gene markers for each cell cycle transition and phase and removes the principal components orthogonal to the cell cycle signatures. The results are in accordance with DeepCycle (Supplementary Fig. S9). The disadvantage of Revelio is the need to define the list of genes related to every phase and so it generalizes less to cell types with different signatures. DeepCycle instead can pre-select the genes showing a cycling signature and exploit them to extract the cell cycle with little previous knowledge.

Regarding the discrete assignment of cells to the different phases, we tested Cyclone[29], while in agreement with DeepCycle's and Revelio's assignments for the ductal cell progenitors and the human fibroblasts, completely fails to detect the correct phases in the mESCs (Supplementary Fig. S9).

DeepCycle produces dynamic trajectories in the unspliced-spliced space for each gene (Fig. 2D) and the quality of the fit of each trajectory to the data can be used to evaluate whether the learning process worked properly. The alternating expression of Cyclins E and B across the transcriptional phase suggests a relation between the transcriptional phase and the cell cycle progression (Fig. 2E). For completeness, examples of noncycling genes are shown in Supplementary Fig. S10.

**Detection of cell cycle phases in multiple cellular models.** Single cells can be associated with the S and G2/M phases by analyzing the expression of representative marker genes[19,30]. The transcriptional phase does not contain information about the cell cycle phase transitions; gene phase markers are needed to identify the different transitions throughout the cell cycle. By integrating the information from S, G2/M marker genes[30] and the number of RNA counts per cell, we devise a strategy to estimate the G1/S, S/G2 and M/G1 transitions (see Methods 'Detection of the cell cycle phase transitions' and Supplementary Fig. S11). Briefly, the G1/S transition corresponds to the peak in cyclin-E1 and -E2, the S/G2 transition to the transcriptional phase where the G2M score increases above the S score, and mitosis to the beginning of the sharp decrease in the RNA counts per cell (Fig. 3A). The cell cycle scores calculated by scVelo match well with the transcriptional phases inferred by DeepCycle (Supplementary Fig. S12). The loss of *Wee1/WEE1*, a protein kinase inhibiting mitosis, allows the cyclin B1-Cdk1 complex to activate the cascade of reactions necessary to proceed into mitosis[31,32] and, consistently, the mRNA levels of the Aurora kinases A (*Aurka/AURKA*) that localize at the centrosomes[33,34], and of the Nucleolar and spindle associated protein 1 (*Nusap1/NUSAP1*), that plays a role in spindle microtubule organization[35,36], increase in G2 and M phases (Fig. 3A). Other possible marker genes show cycling patterns as expected, e.g. *Orc1/ORC1*, *Mcm6/MCM6*, *Ccne1/CCNE1*, *Ccna2/CCNA2*, and *Ccnb2/CCNB2* (Fig. 3A).

To simplify the comparison of the cell cycles across datasets, the transcriptional phases were normalized between 0 and 1 and were aligned such that mitosis occurs at $\theta = 1$. The paths of the cells around the cell cycle can easily be identified in the 2-dimensional projections (Fig. 3B). Though the extended RNA velocity model[19] did not capture the correct dynamics at the level of the single gene (Supplementary Fig. S3), it could infer the correct dynamics of transcriptional changes at the cell level (see the velocity plots in Supplementary Fig. S12). To add a complementary experimental validation independent of the

scRNA-seq, we designed a series of bulk RNA-seq in a cell cycle-sorted population of cells (G1, S, G2/M). The cells from the scRNA-seq sorted by transcriptional phase are close to the correct phase with a slight shift. Notice that the phases defined by FACS and by DeepCycle are based on different markers. In the case of FACS we defined phases according to DNA content. When we analyze scRNA-seq data with DeepCycle we do not have information about DNA content and we use gene markers. Thus, the phase transitions may not align perfectly (Supplementary Fig. S13).

Fast cell cycles are typically associated with pluripotency and stemness[11–13]. Consistently, the mESCs present the lowest number of cells in G1, while fibroblasts and ductal cells have much more extended G1 phases (see Fig. 3). More realistic views of the cell cycle durations can be produced by rescaling all the transcriptional phases to have the same S phase length, as known from the literature the S and M phases are quite constant, constrained by the structural events happening in the cells and do not depend on the different cell types[37] (see Supplementary Fig. S14). The fractions of mESCs assigned to the different phases are 19% to G1, 40% to S, and 41% to G2/M (Fig. 3C). By staining cells with propidium iodide followed by flow cytometry analysis, similar fractions of cells are detected in the main cell cycle phases, respectively, 22–26% in G1, 42–51% in S, and 27–32% in G2/M (Supplementary Fig. S15).

At mitosis, the mother cell needs to have approximately double its original volume in order to generate two daughters of the same initial size. Droplet-based single-cell technologies, such as 10x, can indirectly detect the different cell sizes, where a bigger cell means a higher concentration of mRNA within the droplet, which should reflect a higher count of unique RNA molecules (UMIs) within the cell. In this case, the increase in the unique RNA molecules across the cell cycle should be roughly proportional to 2. Indeed, as predicted, the RNA counts per cell as a function of the transcriptional phase show a positive fold change of 2.1, 2.0, and 2.1 for mESCs, ductal cells, and fibroblasts, respectively (Fig. 3C). Further, the flow cytometry analysis performed for the mESCs showed roughly a doubling size passing from G1 to G2/M phases (Supplementary Fig. S15). After validating that the transcriptional phases identified by DeepCycle are consistent with the global features of the cell cycle, such as cell cycle markers, cell sizes, and fractions of cells in each phase, we can discuss the regulation of individual cell cycle genes at the mRNA level. The fibroblasts show contamination of cells from the nonproliferative subpopulation, arrested in mid-G1, as discussed in the last section.

Members of the Cdc25 family are well conserved key regulators of the cell cycle[38–40]. The mRNA expression of the Cdc25 family of proteins shares the same behaviour across the datasets, i.e. Cdc25a/CDC25A increases at the G1/S transition while Cdc25b-c/CDC25B-C at the G2/M (Supplementary Fig. S16), consistently with the function of their protein products[41].

The minichromosome maintenance protein complex (Mcm) is a heterohexamer, formed by Mcm2-7/MCM2-7 proteins, which works as a helicase that unwinds the double-stranded DNA and powers the replication fork progression during the S phase[42]. As expected, the mRNA levels of all subunits of the Mcm peak at the beginning of the S phase for all the datasets (Supplementary Fig. S16).

*Cdk1/CDK1* mRNA level increases in G2 and M phases as required by its protein function[43] (Fig. 3A). The other main Cdk mRNAs (*Cdk2-4-6/CDK2-4-6*) show lower expression levels across phases and are less consistent across the datasets, they might rather be regulated at the protein level, translationally or post-translationally (Supplementary Fig. S16). It has been previously shown that protein levels of the cyclin-E and A do

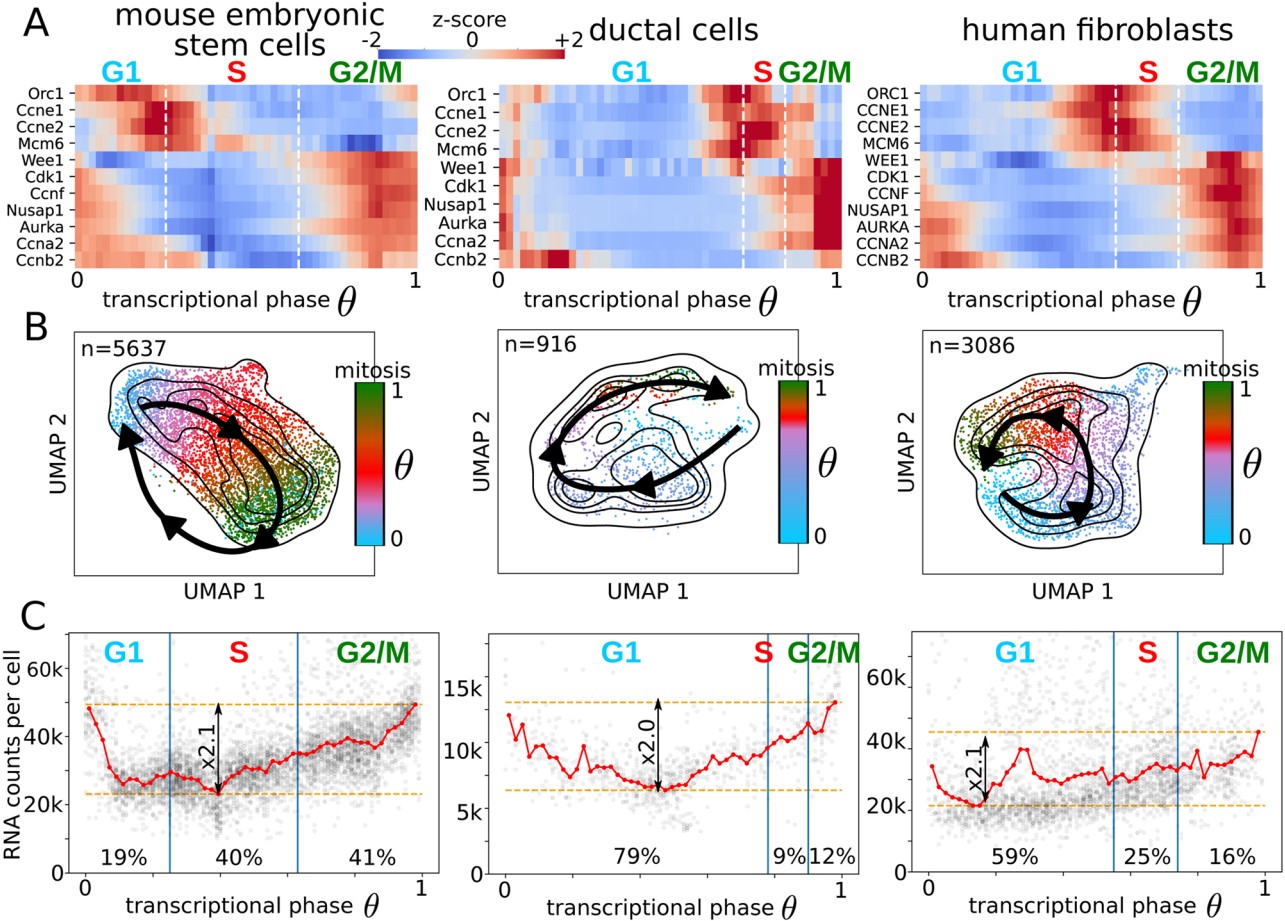

**Fig. 3 Cell cycle analysis in mouse and human cellular models. A** The transcriptional phases and the cell cycle phases were connected based on marker genes. Z-scores are only intended for comparison purposes, the changes in the expression across the cycle are, in general, highly significant. **B** The UMAP embeddings for mESCs, ductal cells, and human fibroblasts with the cell cycle directionality identified by DeepCycle (black arrows). The cells are associated with different colors depending on the cell cycle phase they belong to, light blue for cells in G1, red in S, and dark green in G2/M. **C** RNA counts per cell as a function of the transcriptional phase show doubling trends followed by a sudden drop that identifies mitosis.

not change across the mESC cell cycle[44], but instead, mRNA levels are upregulated at the G1/S transition and in the G2/M phase, respectively (Fig. 3A).

Finally, DeepCycle allows a genome-wide investigation of gene expression dynamics across the cell cycle. Indeed, we observed different waves of gene expression during the different phases of the cell cycle (Supplementary Fig. S17). Overall, DeepCycle consistently identifies cycling genes and shows their mRNA synthesis rate (unspliced) and expression level (spliced) across the cell cycle.

**Prediction of cell-cycle core transcription factors**. Having characterized the cell cycle at the mRNA level, another feature of our approach is that it allows us to identify potential transcription factors (TFs) responsible for gene expression dynamics. Transcription factors bind to DNA-specific sequences (binding motifs) and activate transcription of their target genes. They encode the cellular programs for many of the functions a cell needs to perform. To infer the TFs active during the cell cycle, we implemented an ISMARA-like approach[45]. Briefly, Balwierz et al. introduced a linear model to infer TF activities from bulk RNA-seq samples. To apply it to our data, we used the same linear model to try to explain the expression level of the unspliced reads in single cells. Even if the amount of unspliced reads is much lower compared to the spliced reads (~5-6 times less, Fig. 1E), they remove the effect of mRNA stability, reflecting more closely

the nascent transcription events and, therefore, the effect of transcription factors at the gene promoters.

The motif analysis predicts that most TF activity takes place in the G1 phase when cells need to decide whether to go into another round of replication or to arrest the cycle in order to accomplish a new function (Fig. 4A). Among the most significant activities, Yy1/YY1 targets are upregulated in the G1 phase in all the datasets, suggesting a general role during the cell cycle (Fig. 4A, B). Indeed, Yy1 is known to induce proliferation and maintain pluripotency of mESCs through the BAF complex[46]. Interestingly, Yy1 binds to chromosomes during mitosis[47] and, accordingly, its transcription starts already in the G2/M phase suggesting a pioneering activity at the beginning of a new cycle (Fig. 4B).

For both mouse datasets (mESCs and ductal cells), the E2f family appears as a critical group of regulators. Members of the family are known to act at the beginning of the cell cycle specifically for the G1/S transition and to become active after the phosphorylation of the retinoblastoma proteins (pRb)[1,48,49]. Two E2f-related motif activities (E2f1, E2f2_E2f5) peak in between the G1 and the S phase, presumably to activate the genes necessary for the transition[50] (Fig. 4A). More specifically in mESCs, E2f1 seems to be mostly regulated at the protein level since the change in the mRNA level (~50%) is very little compared to the change in activity (Fig. 4B). Other factors seem to act similarly between mESCs and ductal cells, like the TATA-binding protein-

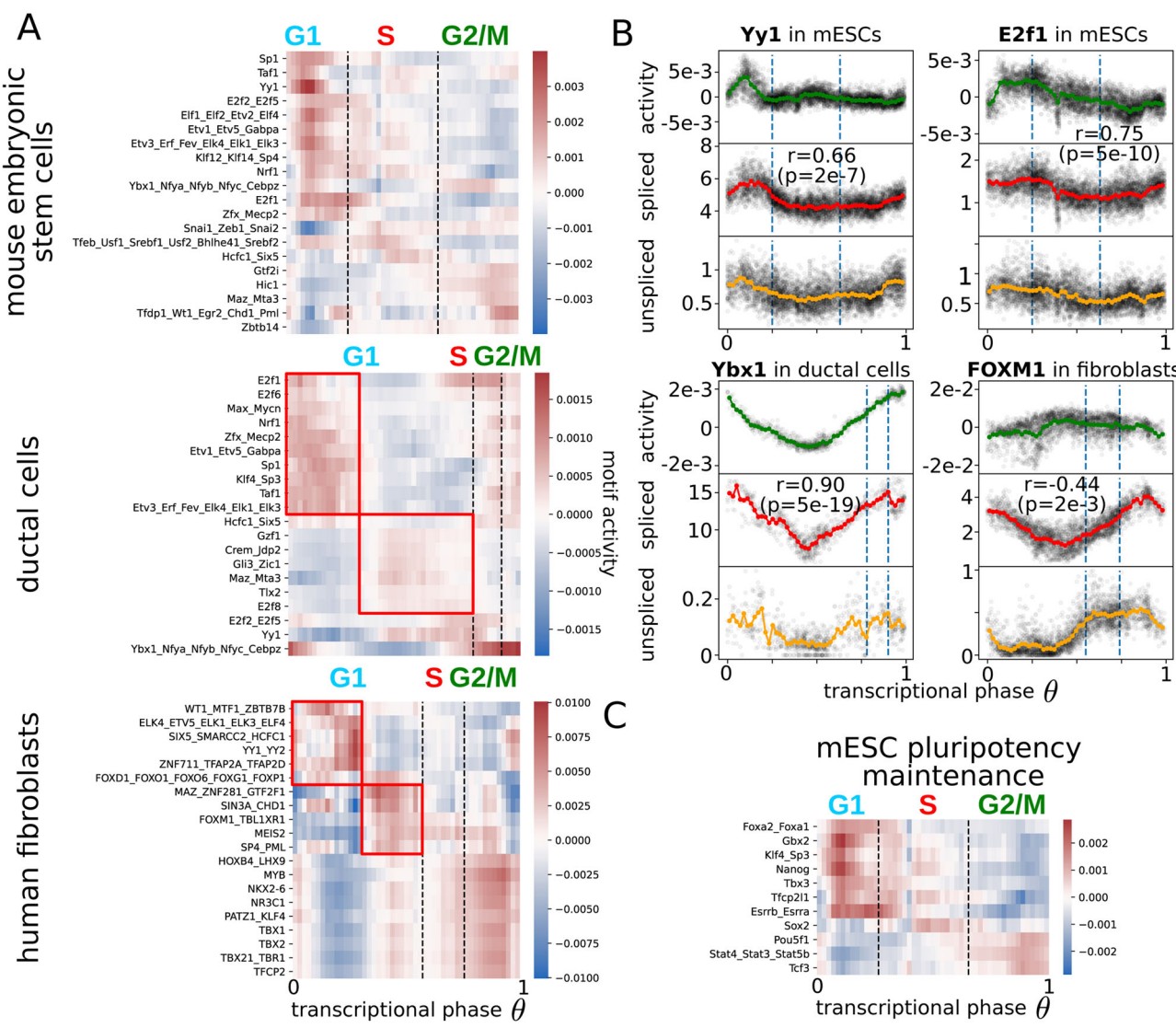

**Fig. 4 Transcription factor dynamics driving expression during the cell cycle. A** Motif activities for families of transcription factors across the cell cycle in mESCs, ductal cells, and human fibroblasts. The red boxes identify two waves of transcription in the G1 phases for more differentiated cell lines. **B** The comparison of motif activities with the respective mRNA levels around the cell cycle for Yy1 and E2f1 in mESCs, Ybx1 for ductal cells, and FOXM1 for human fibroblasts. These comparisons allow clarifying whether the regulation is happening more at the transcriptional level or the protein level. The Pearson's correlation coefficients between spliced reads and motif activities are reported (*r* values) as well as the exact tests of no-correlation (*p* values). **C** Dunn et al.[58] identified two sets of TFs at the core of the pluripotency maintenance network in mESCs, the first responsible for the integration of the signals (input) and the latter for taking a fate decision (computation). The heatmap shows their activities around the cell cycle of mESCs for the pluripotency factors for which the binding motifs are known. The 'input' motifs are active in late G2/M while the 'computation' factors in the early G1.

associated factor (Taf1), the Specificity factor 1 (Sp1), and the Nuclear respiratory factor 1 (Nrf1), all active in early G1 (Fig. 4A). Regarding the ductal cells, we found a very high correlation ($r = 0.91$, $p = 4e-19$ exact test of no-correlation) between Ybx1 mRNA level and the activity of its motif where both are constantly increasing from G1 to M (Fig. 4B). Interestingly, Ybx1 is known to positively regulate the G1 and G2/M phases of the cell cycle[51,52]. Regarding the factors appearing in the human fibroblasts, MYB plays a role in the G2/M transition, with a constant increase of expression from G1 to G2/M[53,54]. Also, its targets follow the same trend, and MAZ induces MYB expression shortly after the exit from quiescence, bypassing the inhibition of E2F-pRB[55] (Fig. 4A). Similar to Ybx1 in ductal cells, the mRNA level of FOXM1 grows constantly from G1 to M as expected by its function during mitosis[56,57], but the activity of its targets is slightly anticorrelated, hinting at a complex post-transcriptional regulation[57].

For mESCs the maintenance of the pluripotent state is crucial and the main factors involved in the pluripotency transcriptional program are known[58] (Fig. 4C). Among them, the strongest activation happens for the targets of Stat3[59]/Stat4/Stat5b, Tcf3, and Pou5f1 (Oct4), which are increased in G2/M, followed by Klf4[60]/Sp3, Gbx2, Nanog, Tfcp2l1, and Essrb[61,62]/Essra in G1 (Fig. 4C).

From a general perspective, a clear pattern emerges by comparing the undifferentiated mESCs with the more differentiated human fibroblasts and ductal cells. The undifferentiated cells show a strong and unique wave of activation of TFs in G1. Instead, in the more differentiated cell types, the activities of the TFs across the G1 phase cluster into two groups. The first group displays an early activation directly after mitosis, while the second group exhibits a late G1 activation (red boxes in Fig. 4A). We believe these waves are linked to cell-fate decisions, as discussed in the next section.

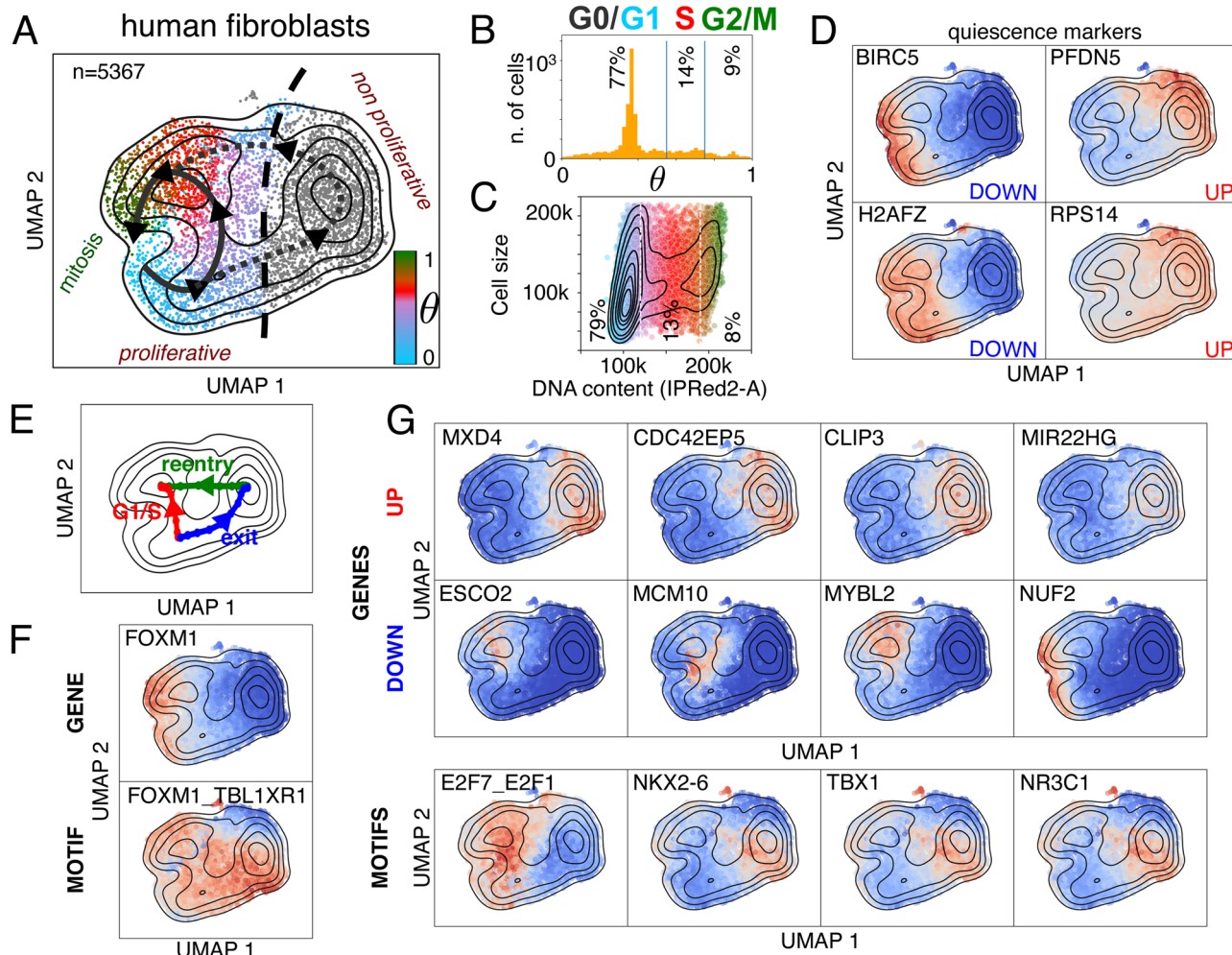

**Fig. 5 G1/S transition and cell-cycle exit in human fibroblasts. A** The two subpopulations of human fibroblasts, separated by the black dashed line: on the left side the proliferative cells, and on the right the nonproliferative. Each cell is colored according to its inferred transcriptional phase. **B** The distribution of transcriptional phases for the human fibroblasts shows the nonproliferative fibroblasts are closer to the cell cycle stage associated to $\theta = 0.3–0.4$ (mid-G1). **C** Flow cytometry analysis recapitulates fractions of cells in the main phases of the cell cycle similar to the phases identified with DeepCycle. The cells in G0/G1 can be bigger than the cells in G2/M. The colors are consistent with panel A, G1 in blue, S in red, and G2M in green. **D** Examples of G0 marker expressions for the two subpopulations[64]. BIRC5 is also known as survivin. Color intensity reprsents gene expression upregulated level (red) and downregulated level (blue). **E** The paths, going from mid-G1 towards the S phase (in red and labeled as 'G1/S'), from the mid-G1 phase towards the nonproliferative state (in blue and labeled as 'exit'), and the nonproliferative state toward the S phase (in green and labeled as 'reentry'), are identified by the higher density of cells. **F** FOXM1 is downregulated in the nonproliferative cells and shows the opposite trend along the G1/S and exit paths (red and blue paths in panel E). The activity of its motif is completely uncorrelated with its expression. **G** The top up- and down-regulated genes with the strongest fold changes, compare the paths in **E**. The third row shows the most significant motif activities identified by comparing the paths in **E**. Color intensity reprsents either gene expression or motif activity (red: upregulation; blue: downregulation).

**Characterization of cycling cells shifting to a cycle-arrested state**. The human fibroblasts include a subpopulation with a low cycling activity, which was excluded from the previous analysis (Supplementary Figs. S1-S2). By mapping the 'nonproliferative' cells across the cycle with the model trained with DeepCycle, this sub-population was closer to cycling cells associated with the mid-G1 phase (peak in Fig. 5B). Therefore, the full population of fibroblasts comprises 76% in G0/G1, 15% in S, and 9% in G2/M phases (Fig. 5B). Similar numbers have been obtained through flow cytometry analysis, where the DNA content assigns 79% to G0/G1, 13% to S, and 8% to G2/M phases (Fig. 5C). Further, the flow cytometry analysis shows that some cells in G1 can be as big as cells in S and G2/M suggesting that the cells waiting in G0/G1 are increasing in size; it remains unclear whether they will re-enter the cycle later. The cell velocities are consistent with our

interpretation and do not clarify if the cells in the alternative state will start cycling again (Supplementary Fig. S18).

The two subpopulations split their trajectories around mid-G1 (Fig. 5A) and specific markers of quiescence[63,64] suggest the nonproliferative cluster might include cells transitioning into the G0 phase (Fig. 5D and Supplementary Fig. S19-22). To detect the underlying changes in the gene expressions and their regulations, we implemented a method based on a modified version of the Nudged Elastic Band[65], to infer the paths connecting the cell states in the bidimensional space (Fig. 5E). The detected paths follow the trajectories with the highest density of cells, as shown in Fig. 5E.

To strengthen our hypothesis of the nonproliferative cells being at an early stage of quiescence, we checked the expression along the two paths of G0 markers[63,64]. The markers supposed to be

G0-downregulated are consistently inactivated (Supplementary Figs. S19-S20) and similarly activated for G0-upregulated genes (Supplementary Figs. S21-S22). Overall, we cannot exclude that nonproliferative fibroblasts might represent a differentiated state of the fibroblasts and not simply reflect cells entering in quiescence[66]. FOXM1 is strongly downregulated in nonprolifera-tive cells, and it is indeed a known marker of quiescence[63]. With regards to the cell cycle analysis (Fig. 4B), FOXM1 targets do not follow its mRNA expression but are still upregulated while exiting the cell cycle (Fig. 5F). The proliferation-quiescence decision is controlled by a bifurcation in CDK2 activity[67], which is consistent with the expression of CDK2 mRNA in the population of fibroblasts (Supplementary Fig. S23). Among the top genes upregulated along the path toward the nonproliferative state (the blue and green paths vs the red path in Fig. 5E), we find MXD4, CDC42EP5, CLIP3, and MIR22HG. MXD4 is a MYC antagonist known to increase the fraction of cells in the G0/G1 phase in hematopoietic differentiation[68], and could be a master regulator of entry into the quiescence-like state. CDC42EP5 is a small Rho-GTPase belonging to the Borg family and is involved in cell shape regulation and lamellipodia formation[69]. Similarly, CLIP3 (or CLIPR-59) is a CAP-Gly domain-containing linker protein with a poorly-specified function, perhaps modulating the compartmen-talization of the AKT kinase family[70]. Lastly, MIR22HG is a long non-coding RNA involved in proliferation that acts as a tumor suppressor in primary lung tumors[71] and leads to poor prognosis in glioblastoma[72]. On the other side, among the most down-regulated genes in the quiescent state are ESCO2, MCM10, MYBL2, and NUF2. ESCO2 is needed during the S phase to modify cohesin[73] and MCM10 accumulates during the S phase while being lowly expressed during the rest of the cycle[74]. MYBL2 (B-Myb) belongs to the family of the MYB transcription factors and has been typically associated with poor prognosis in cancer[75], while NUF2 localizes at centrosomes and is necessary for mitotic progression in vertebrates[76,77]. For the extended lists of up and down-regulated genes see Supplementary Figs. S24-S27.

Importantly, the top motif, that distinguishes the two subpopulations (E2F7_E2F1), belongs to the E2F family, which is one of the master regulators of the cell cycle[1], and is strongly inactivated in the non-proliferating quiescent cells. The other TFs shown in Fig. 5G (NKX2-6, TBX1, and NR3C1) do not have a clear function associated with the cell cycle, so further studies are needed to elucidate their role. More TF motifs associated with the paths are shown in the Supplementary Figs. S28-S29.

In summary, DeepCycle allowed us to characterize G0 transitioning cells in wild-type fibroblasts without having to perturb the cells, finding previously unknown candidate genes and transcription factors regulating quiescence.

## Discussion
We generated scRNA-seq datasets in mouse embryonic stem cells and human fibroblasts with high sequencing depth. The circular RNA velocity patterns emerged clearly in cell-cycle regulated genes revealing the activation/inactivation phases that these genes undergo during the cell cycle. We developed DeepCycle, a deep learning approach, to exploit the RNA velocity patterns and study gene regulation dynamics during the cell cycle. DeepCycle assigns a cell-cycle transcriptional phase for each cell by fitting the RNA velocity patterns. Furthermore, the inferred transcriptional phase can be associated with cell-cycle phases thanks to known gene markers. Thus, DeepCycle allows us to determine the cell-cycle progression state of each cell from scRNA-seq data and identify genes involved in the cell cycle. Importantly, the efficacy of the method was extensively proven in cellular models from different organisms at different developmental stages. Given the variability

in the cell cycle signatures among cellular models, defining the cell cycle phases in RNA data based solely on gene markers lacks generalizability. In the future, the usage of gene markers needs to be replaced by the adoption of methods relying on dynamical features of gene expression, able to accommodate changes in the regulation of the cell cycle.

The decision to implement this approach came after noticing the failure of the current methods within the RNA velocity framework[7,19], to correctly infer the dynamics of the cycling genes. DeepCycle's ability to infer cycling patterns in the spliced-unspliced RNA space at the gene level shows that the framework of the RNA velocity can be further improved by the study of more flexible models of transcription. Likely the assumptions in the previous model (constant rates) should be relaxed to fit the transcriptional model to the data. It is reasonable to imagine that the transcription, splicing, and degradation rates are complex functions changing during the cell cycle progression. Our method will allow the analysis of trajectories without making assumptions about the model parameters, enabling more focus on the dynamics of the single gene.

Furthermore, we envision extending DeepCycle as a Varia-tional Autoencoder (VAE), a neural network capable of model-ling distributions over the input data. VAEs have already been applied to scRNA-seq data as imputation methods to correct for capture rate and noise[21,78–80]. In our case, it will allow us to learn the posterior distribution of the transcriptional phase and model the whole distribution of unspliced-spliced RNAs.

The analysis highlighted known and unknown cell cycle reg-ulators in established cell lines, identifying two major waves of transcription in the G1 phase of differentiated cells while plur-ipotent cells seem to undergo a single wave of transcription during G1. The two waves are likely to be associated with the restriction point where the cells finally commit to undergoing another cell cycle. Further, for the first time, we could observe single cells exiting from the cell cycle in a scRNA-seq sample and disentangle the underlying regulations, thereby providing lists of targets for the regulation of the cell cycle and the quiescent states in mammalian cells. We envision that our approach will facilitate the characterization of the branching point between the S and G0 phases in multiple cellular models by applying it to other scRNA-seq datasets. In particular, an extensive study of the transcrip-tional changes happening at the cell cycle while cells reach con-fluence is still missing and of general interest.

Finally, we anticipate that DeepCycle will become an essential tool for the scientific community to further investigate the cell cycle in a broad range of systems without the need for cell syn-chronization or genetic-tagging and complement the experi-mental methods that have been used in the past to unravel the regulation of the cell cycle[67,81,82]. This makes our approach especially suitable to study the interplay of the cell cycle with pluripotency and cell reprogramming[83]. Moreover, the compar-ison between normal and cancer tissues may lead to the discovery of cell-cycle dysregulated mechanisms in tumors and, perhaps, potential targets for drug development.

## Methods
**Cell culture**. E14Tg2a.4 mouse embryonic stem cells (ECACC General Cell Col-lection; catalogue number: 08021401) were cultured on 0.1% gelatin-coated culture plates in DMEM (4,5 g/l glucose) supplemented with GLUTAMAX-I, 15% heat-inactivated fetal calf serum (42F5874K, ESC culture tested, GIBCO), 0.1 mM beta-mercaptoethanol, 0.1 mM nonessential amino acids 1500 U/ml leukemia inhibitory factor (produced in house), 3 μM CHIR99021 (72054, Stem Cell Technologies) and 1 μM PD0325901 (72184, Stem Cell Technologies) in 5% $CO_2$ at 37 °C.

IMR90 primary human fetal lung fibroblast cells (CORIELL Institute for Medical Research, Reference: I90-19) were cultured in DMEM 41966 (4,5 g/l glucose) supplemented with 10% fetal calf serum, Penicillin 100 UI/ml, and Streptomycin 100 μg/ml in 5% CO2 at 37 °C. The cells were at passage 21 when

performing the experiments. For both scRNAseq and FACS experiments, 20,000 cells per well were seeded into 6 well plates and cultured for 72 h.

**Single-cell RNA sequencing**. To obtain single-cell suspension of mESCs for single-cell RNA sequencing, cells on a 60 mm culture dish were washed once with PBS and treated with 1 ml 0,25% trypsin-1mM EDTA (25200-072, Invitrogen) at 37 °C for 3 min, then harvested into 3 ml medium containing serum, and washed 2-times with PBS containing 0.04% BSA. To prepare single-cell suspension of IMR90 cells, the cells from one well of a 6 well culture plate were washed twice with PBS and treated with 500 ul 0.05% trypsin-0.53 mM EDTA (25300-062, Invitrogen) at 37 °C for 2 min, then harvested into 4.5 ml medium containing serum, passed through 50 µm cell strainer and washed 2-times with PBS containing 0.04% BSA. In both cases, cell concentration and viability (98%) was determined using Countess II (Invitrogen) according to the manufacturer's instructions. Cells were then processed using the 10x Genomics Chromium System according to the manufacturer's instructions.

Cell number and viability were determined by a Trypan Blue exclusion assay on a Neubauer Chamber. Samples consisting of >90 percent viable cells were processed on the Chromium Controller from 10x Genomics (Leiden, The Netherlands). Ten thousand cells were loaded per well to yield approximately 6500 captured cells into nanoliter-scale Gel Beads-in-Emulsion (GEMs).

In the case of mESCs, the single-cell 3 prime mRNA seq library was generated according to 10X Genomics User Guide Chromium Single Cell 3′ Reagent Kits v3 (P/N CG000183 Rev A). For the human fibroblasts, the single-cell 3 prime mRNA seq library was generated according to 10x Genomics User Guide Chromium NEXT GEM Single Cell 3' Reagent Kits v3.1 (P/N CG000204 Rev D). The raw and processed data for both libraries were stored on the GEO (Accession number: GSE167609).

CellRanger outputs have been processed with velocyto[9] (version 0.17.17) and analyzed using scanpy[84] (version 1.4.4.post1) and scvelo[19] (version 0.2.2) and imputed spliced and unspliced reads from scvelo.pp.moments have been used for the analysis.

**Cell cycle assay and flow cytometry**. Cells were harvested by trypsin as before and washed once with PBS. About $2 \times 10^6$ cells were resuspended in 100 µl PBS and added drop-by-drop to 900 µl 95 % ethanol, while mixing, then stored at $+4$ °C overnight. Cells were then collected by centrifugation, washed once with PBS, resuspended in 1 ml staining buffer (50 µg/ml propidium iodide, 2 mM MgCl₂, 50 ng/ml RNaseA [EN0531, ThermoScientific] in PBS) and incubated for 20 min at 37 °C. Stained cells were washed once with PBS and analyzed on BD LSRII flow cytometer.

The fcs files were processed with fcsparser (https://github.com/eyurtsev/fcsparser). For the mESCs, the debris in the data was removed by filtering SSC-H and SSC-W values higher than 140,000 and 100,000, respectively, and by selecting cells with a Hotelling $T^2$ value lower than 6 in the FSC-A SSC-A space, see Supplementary Fig. S15. The filtering retained more than 80% of the original cells (~26k out of 31k). For the human fibroblasts, SSC-H values lower than 25,000 and SSC-H and SSC-W values greater than 150,000 and 110,000, respectively, were excluded. As for the mESCs, only cells with Hotelling $T^2$ lower than 6 in the FSC-A SSC-A space are retained.

For bulk RNA sequencing of isolated cells of different cell cycle phases, $5 \times 10^6$ E14Tg2a cells were stained in FACS tubes in 5 ml culture medium for 30 min at 37 °C with Vybrant DyeCycle Violet (V35003 Invitrogen). Cells were then harvested by centrifugation at 1000 rpm (61x $g$) for 4 min, resuspended in 500 µl medium containing 50 nM TOPRO-3 (T3605 Invitrogen) and sorted on BD FACS ARIA II. First, TOPRO-3 negative live single cells were gated, then G0/G1, S and G2/M phases were sorted based on the distribution of the Vybrant DyeCycle Violet signal (Supplementary Fig. S13A). Sort purity was verified by analysing a small aliquot of the sorted cells.

**Implementation of DeepCycle**. The autoencoder was implemented in TensorFlow 2. The input and output layers of the autoencoder consist of Densely connected layers of size twice the number of input genes to accommodate spliced and unspliced read values. The subset of genes in the GOterm:cell_cycle (GO:0007049) passing through the hotelling filter (see section 'Identification of cycling genes and high-density paths') has been fed to the autoencoder. The detailed structure of the autoencoder is depicted in Supplementary Fig. S4. The Densely connected layers in the blue boxes have a size equal to 4 times the number of genes and are activated through a leaky ReLU function. The neural network in the orange box calculates the atan2 for the gene selected as the input gene and concatenates this value with the output of the dense layers from the first part of the encoder (blue box). The concatenation is fed to a Dense layer of size four times the number of genes and outputs a real number (θ). The real number is the input of the decoder that transforms it to (cos(θ), sin(θ)) with the layer Circularize. The bidimensional vector is then fed to a series of densely connected layers till the output layer. The GaussianNoise layers add gaussian noise to the inputs to avoid the neural network overfitting the data.

The training is performed in 2 steps: 1) training the encoder on the phases estimated from the input gene (atan2 of z-scored spliced and unspliced reads),

*Nusap1* for mESCs, *Ccnd3* for ductal cells, and *MELK* for human fibroblasts; 2) training encoder+decoder to reconstruct the unspliced-spliced reads. Both training steps have an early stop when they reach a plateau

*tf.keras.callbacks.EarlyStopping(monitor = 'val_loss', min_delta = 0.0, patience = 20, verbose = 1, mode = 'auto', restore_best_weights = True)* and the learning rate decreases accordingly with *tf.keras.callbacks.ReduceLROnPlateau(monitor = 'val_loss', factor = 0.8, patience = 5, min_lr = 0.00001)*. 17% of the input cells are used as validation and the training is performed in batches of 5 cells. SGD, RMSprop and Adam optimizers have been tested and the latter (Adam) was the one giving the best performance. The optimization has been performed on the loss function Mean Squared Error (MSE) between the input and the output.

Finally, to infer a phase for each cell, we binned the angles in 50 and assign a cell to the closest bin in the unspliced-spliced space predicted by the autoencoder (red lines in Fig. 2C, D and black in Supplementary Fig. S5) for all the genes used for the training (GO term: cell_cycle, GO:0007049).

DeepCycle implementation was stored in the GitHub repository https://github.com/andreariba/DeepCycle.

**Detection of the cell cycle phase transitions**. The transcriptional phase contains only the information about the succession of states composing the cell cycle. To annotate the different cell cycle phases, we analyze the expression of cell cycle markers[30] and the number of RNA counts per cell. We computed 2 scores based on the marker genes in the S and G2M phases[30]. For all the expressed genes in the two lists we computed the z-score expression across all the cells and then calculated an average z-score per cell for all the genes. Similarly, we computed a CyclinE score by considering only Ccne1-2/CCNE1-2 genes. Finally the transitions are defined as follows.

- M/G1: the first bin across the transcriptional phase when the number of RNA counts (UMI) per cell drops;
- G1/S: the first bin across the transcriptional phase after the CyclinE score reaches its maximum;
- S/G2: the theta where the G2M score becomes greater than the S score.

The detailed analysis for the three datasets is shown in Supplementary Fig. S11.

**Validation bulk RNA-seq in cell cycle-sorted populations of cells**. The three samples were sorted as in section 'Cell cycle assay and flow cytometry' and the libraries have been prepared as follows. Total RNA from $5 \times 10^5$ cells in G0/G1, S and G2/M phases was extracted using the Machery-Nagel NucleoSpin RNA kit (740955.50) according to the manufacturer's instructions. Total RNA-Seq libraries were generated from 500 ng of total RNA using TruSeq Stranded Total RNA Library Prep Gold kit and TruSeq RNA Single Indexes kits A and B (Illumina, San Diego, CA), according to manufacturer's instructions. Briefly, cytoplasmic and mitochondrial ribosomal RNA (rRNA) was removed using biotinylated, target-specific oligos combined with Ribo-Zero rRNA removal beads. Following purification, the depleted RNA was fragmented into small pieces using divalent cations at 94oC for 8 minutes. Cleaved RNA fragments were then copied into first-strand cDNA using reverse transcriptase and random primers followed by second-strand cDNA synthesis using DNA Polymerase I and RNase H. Strand specificity was achieved by replacing dTTP with dUTP during second-strand synthesis. The double-stranded cDNA fragments were blunted using T4 DNA polymerase, Klenow DNA polymerase and T4 PNK. A single 'A' nucleotide was added to the 3' ends of the blunt DNA fragments using a Klenow fragment (3' to 5'exo minus) enzyme. The cDNA fragments were ligated to double-stranded adapters using T4 DNA Ligase. The ligated products were enriched by PCR amplification (30 sec at 98 °C; [10 sec at 98 °C, 30 sec at 60 °C, 30 sec at 72 °C] x 12 cycles; 5 min at 72 °C). Surplus PCR primers were further removed by purification using AMPure XP beads (Beckman-Coulter, Villepinte, France) and the final cDNA libraries were checked for quality and quantified using capillary electrophoresis. Libraries were then sequenced on Illumina HiSeq 4000 as 50 bases single-end reads.

Reads were preprocessed in order to remove the adapter, polyA, and low-quality sequences (Phred quality score below 20). After this preprocessing, reads shorter than 40 bases were discarded for further analysis. These preprocessing steps were performed using cutadapt version 1.10. Reads were mapped to rRNA sequences using bowtie version 2.2.8, and reads mapping to rRNA sequences were removed for further analysis.

**Transcription factor activity**. The linear model used to infer the motif activities was implemented as in ISMARA[45]. To find the regulatory interactions between transcription factors and genes, we used Motevo predictions of binding sites in promoters downloaded from the Swiss Regulon Portal (https://swissregulon.unibas.ch/sr/downloads) for mm10 mouse genome assembly (https://swissregulon.unibas.ch/data/mm10_f5/mm10_sites_v2.gff.gz) and hg19 human genome assembly (https://swissregulon.unibas.ch/data/hg19_f5/hg19_sites_v2.gff.gz). Briefly, the Motevo algorithm uses a Bayesian framework to estimate the posterior probability that a binding site for a given weight matrix (associated with a motif) occurs in an interval[85]. After, we summarized the transcription factor binding sites in a matrix of site-counts $N_{pm}$ by summing the posterior probabilities for each motif $m$ in a promoter $p$. We defined a promoter as the TSS $+/-$ 1 kb.

The cross-validation was repeated 10 times and the average optimal strength of the ridge regularization was used for the final calculation of the TF activities.

**Identification of cycling genes and high-density paths**. A mixture of two bivariate Gaussians was used to fit the distribution of unspliced-spliced expressions, to identify genes showing at least two maxima in the distribution of cells. After identifying the two Gaussians, a Hotelling's $T^2$ test was applied to select the genes with two significantly different attractors. After supervised filtering of the remaining genes, we implemented a method to select genes with at least two paths connecting the two maxima. The path detection was implemented into two steps. First, coarse-grained paths were drawn by slicing the spliced-unspliced landscape and connecting the minima found across the successive slices. To refine the identified paths we implemented the Nudged Elastic Band[65] with the addition of a viscosity term to stabilize the dynamics, we called the method Viscous Nudged Elastic Band (VNEB).

The VNEB was also applied to the fibroblasts dataset to identify the paths connecting the G1 phase to the S and G0 phases in Fig. 5E.

**Reporting summary**. Further information on research design is available in the Nature Research Reporting Summary linked to this article.

## Data availability

The raw data generated in this study has been deposited in the GEO database under accession code GSE167609. The binding site predictions used in this study are avilable in the SwissRegulon database (https://swissregulon.unibas.ch/sr/downloads). The source data files with the results of this paper are openly accessible in Zenodo under accession code 4719436[86].

## Code availability

The code of DeepCycle can be downloaded from the GitHub repository https://github.com/andreariba/DeepCycle. Data analysis was perfomed using python 3.7.9 and the following pyhton packages: scipy 1.5.2, numpy 1.19.1, pandas 1.1.1, scikit-learn 0.23.2, tensorflow 2.2.0, anndata 0.7.4, matplotlib 3.3.1, seaborn 0.10.1, scanpy 1.4.4.post1, scvelo 0.2.2 and velocyto 0.17.17.

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

## Acknowledgements
We thank the cell culture service, the flow cytometry service and the imaging center of the Institut de Génétique et de Biologie Moléculaire et Cellulaire (IGBMC). We thank specially the national platform GenomEast for the sequencing and the 10x Genomics reagents. Special thanks to Deepak Alapatt for helping during the implementation of DeepCycle and Alastair McEwen for carefully reading the manuscript. This study was supported by funds from Conseil National de la Recherche Scientifique, Institut National de la Santé et de la Rechrche Médicale and Université de Strasbourg; the grant ANR-10-LABX-0030-INRT, a French State fund managed by the Agence Nationale de la Recherche under the frame program Investissements d'Avenir ANR-10-IDEX-0002-02 (NM); the fellowship «IDEX chaires attractivité recherche» granted by the University of Strasbourg (NM); and, the USIAS fellowship granted by the University of Strasbourg (NM). Work in the lab of WMK was funded in part by grants from: La Fondation Recherche Medicale (FRM) (AJE20160635985), La Fondation Schlumberger pour l'Education et la Recherche, FSER 19 (Year 2018)/FRM and La Ligue Contre le Cancer.

## Author contributions
A.R., A.O., N.M. designed the research. A.R. and N.M. designed the method. AR implemented the method. A.O., M.D., V.A., and M.C. performed wet-lab experiments to generate the sequencing data. A.R., A.O., and M.D. performed the flow cytometry analysis. A.R., M.J., and C.K. performed the bioinformatic analysis of the sequencing data. A.R. and S.J. performed the motif analysis. A.R. and N.M. wrote the paper. A.O., M.D., S.J. and W.M.K. critically read and edited the paper. N.M. and W.M.K. conducted supervision and obtained grants to fund the research.

## Competing interests
The authors declare no competing interests.
