## [Peer Review File · Nature Communications]

Reviewers' Comments:

Reviewer #1:

Remarks to the Author:

In this manuscript, the authors generated in-house single-cell RNA-seq data in mouse ESC and human fibroblast in order to characterize cell cycle dynamic. To analyze the data, the authors propose using an autoencoder approach called DeepCycle to compress the high-dimensional input (spliced UMI + unspliced UMI per gene for 2N input size) into a one-dimension manifold theta.

Strength of the paper:

- Overall, the paper is well written and easy to follow for general audience
- Authors have done extensively exploratory analysis by sorting cells based on theta and observe cells are mainly sorted by their cell cycle phase, which is biologically interesting.
- All figures are of high quality and nicely displayed with clear labels

Main comments:

1. The authors should tune down the language of novel method because the method is essentially a straight application of autoencoder, and no obvious methodological innovation is made.
2. If authors do want to claim method novelty, quantitative evaluation such as ablation analysis is needed. In particular, how does DeepCycle perform when only spliced or unspliced read (but not both) are provided in terms of recapitulating known cell phase cycle?
3. Only one method called Cyclum is compared with DeepCycle and showed poor UMAP clustering (Suppl. Fig. 6). How does DeepCycle (which is just an autoencoder) compare with other approaches such as scVI, scVI-LD, scGAN, scETM, which properly takes into account distribution of the latent encoded variable in a variational autoencoder framework while addressing batch effects in different ways.
4. How was the autoencoder network architecture derived? Was a validation set used here in terms of reconstruction loss?
5. Given known gene markers, can authors show the model can predict accurately the cell cycle phase in a classification tasks using standard quantitative metric such as prediction accuracy? This may help others to annotate their single-cell data with unknown cell cycle phases.

Reference:

1. Lopez, R., Regier, J., Cole, M. B., Jordan, M. I. & Yosef, N. Deep generative modeling for single-cell transcriptomics. *Nature Methods* 15, 1–11 (2018).
2. Svensson, V., Gayoso, A., Yosef, N. & Pachter, L. Interpretable factor models of single-cell RNA-seq via variational autoencoders. *Bioinformatics (Oxford, England)* 36, 3418–3421 (2020).
3. Bahrami, M. et al. Deep feature extraction of single-cell transcriptomes by generative adversarial network. 570, 1–21 (2020).
4. Zhao, Y., Cai, H., Zhang, Z., Tang, J. & Li, Y. Learning interpretable cellular and gene signature embeddings from single-cell transcriptomic data. *bioRxiv* 2021.01.13.426593 (2021).

Reviewer #2:

Remarks to the Author:

The manuscript 'Cell cycle gene regulation dynamics revealed by RNA velocity and deep-learning' describes a deep learning-based algorithm (Deep Cycle) to predict cell cycle state from scRNA-seq velocity data. The algorithm can be used to analyze genes for their cell cycle involvement once fit, including TFs and potential novel cyclce-regulated genes.

While not being a cell cycle specialist, I think the validation experiments and biological insights gained with Deep Cycle are compelling. The authors provide good evidence that their algorithm outperforms other existing tools in cell cycle fitting. The paper is written in good English.

While I think the biological part is convincing (while not being an expert) I think some other parts could improve or have shortcomings.

Major points

1) I think the technical sections are way too short to be understandable. In other words, I think the description of the model and training of it are insufficient. Layer sizes, primary training, why two training steps? How do you map from vectors to an angle, do you make sure to keep unit length vectors through layer norm? What optimizer, loss function, when do you stop training? None of this information is supplied in the git page, one would have to dig deeply into the code to figure everything out.

2) When I read up on Cyclum (to understand how different their algorithm might be to Deep Cycle) I came to understand that most of Deep Cycle seems to be based on Cyclum. Again, this is a bit tricky to judge as there is hardly any information on the Deep Cycle algorithm but suffice it to say that I started to understand how Deep Cycle might work by reading the Cyclum paper. The main difference seems to be that Deep Cycle uses RNA velocity information as input from scVelo, Cyclum does not. In other words, Deep Cycle's algorithmic novelty seems incremental, at best. It bothered me that the authors don't clearly state that their 'algorithm' is heavily based on Cyclum, while I have to admit that it is hard to judge, due to the extremely limited technical information given in this paper.

Minor points

1) To me, it is really hard to understand what exactly this paper is about purely from the abstract. It is very high-level claims and no examples or concrete results. I would definitely rewrite this a bit.

2) Many figures have very short legends, to the degree that it is virtually impossible to understand what is shown in the figures by just reading the legends. Examples are: Figure S3 - no details on the three inputs, no details on the sizes of the layers (no activation information), no info on the sigma layer; Figure S6 - no mention of A and B, no description of what x and y are, no mention of which data was used. These are just two examples but this lack of description is pervasive.

3) The introduction is incoherent and not complete. The authors talk early on about the 'unspliced-spliced RNA space' without ever mentioning what this is and how this is relevant to cell cycle prediction. They never mention the RNA velocity theory in the introduction, which is the basis of this paper. When they mention the theory in the results section, they don't explain what it is (I had to piece it together from the text).

Stefan Bonn

Reviewer #3:

Remarks to the Author:

Riba et al. proposed DeepCycle, an autoencoder model which uses a single latent variable to describe the cell cycle status. DeepCycle is conceptually novel and might potentially provide great biological insights. The DeepCycle algorithm per se was well described and the corresponding benchmarking analysis was rigorous. However, I think further clarification is needed for the biological applications, especially in sections "identification of cell-cycle core transcription factors" and "characterization of cycling cells shifting to the cycle-arrested state", to make the study scientifically sound. I thus have the following comments:

1. Throughout the paper, the dot size in scatterplots might be too big. I fully acknowledge that the authors used density plots to describe the distribution of cells, but it would be better if the authors could also provide a clear representation of individual cells.

2. For Figure 1D, the scrutiny of data quality is highly appreciated. Based on my experience, in general 1) primary cells have less mRNA compared to cultured cells, 2) human cells have more mRNA compared to mouse cells, and 3) "stem-like" cells have more mRNA compared to "differentiated" cells. So the #UMIs per cell differences might be a piece of "real biology", rather than only a quality issue. Since they authors also provided spliced/unspliced% reads in Figure 1E so I'm convinced about the quality of the datasets used here.

3. For Figure 1F, are the two axes represent the fraction of spliced/unspliced reads? Or some integrated "velocity scores" from the spliced-unspliced space? A short explanation would be really appreciated. The same comment applies to other "velocity plots".
4. For citations 21-28, considering DeepCycle is based on AE, I would suggest the authors only cite other algorithms based on AE, since VAE is generative model and might not be directly comparable.
5. For Figure 2C, The "un" represents abundance of unspliced gene #n, and the "sn" represents abundance of spliced gene #n, correct? I think the authors should clarify the two terms in the figure caption.
6. For Figure 2C, the red circle was constructed from the theta angle latent variable, correct? Then my question is how the authors correspond sin/cos theta trajectory on the "velocity space".
7. For Supplementary Figure S3, the decoder takes u and s to calculate the Atan2 , meanwhile imputes Gaussian noise to the data to increase the robustness, correct? As for the decoder, it recapitulates the expression matrix by "circularization" from theta, correct? I think the authors should provide a more detailed explanation to the framework. Some intuitions on the design strategy of the framework would be highly appreciated.
8. For "GOterm:cell_cycle", have the authors tried using all genes? The reason I'm asking is that the scLVM paper (<https://www.nature.com/articles/nbt.3102>) claimed other genes might also be affected by cell cycle. Further, maybe extending the analysis to all genes will introduce additional biological features besides cell cycle in the latent variable, especially when analyzing multiple cell types together? Could the authors please comment?
9. For Supplementary Figure S4, I think the authors should specify which dataset is presented in the figure.
10. For Supplementary Figure S5, I would suggest the authors to describe how the "cell cycle score" represented by the x-axis is calculated. Also, I would suggest the authors to use density plot here, considering there is an aggregation of dots at the top-left corner. The authors should confirm that most of the dots are on the diagonal.
11. For Supplementary Figure S6, again, I think the authors should specify which dataset is presented in the figure.
12. For Supplementary Figure S7, are the authors using the same UMAP layouts as Figure 3A and Supplementary Figure S8? Seems the same layouts are used so the claim here is valid. I would suggest the authors to put the UMAP plots color-coded by scVelo scores and DeepCycle theta values side-by-side to better support the claim here.
13. For the flow cytometry analysis, I really appreciate the experimental validation here. However, it seems that the authors didn't describe the cell staining strategy. Was Hoechst staining performed to determine cell cycle stages?
14. For section "identification of cell-cycle core transcription factors" in general, I think the correct way of demonstrating "identification" would be 1) identify top TFs that are correlated with theta, and 2) show these top TFs are biological meaningful. So I think the authors should either 1) perform a real "identification" analysis, or 2) only claim they could recapitulate known biology as an additional support for validating the theta value.
15. Specifically, for Figure 4B, are the r values calculated against the abundance of all, spliced or unspliced mRNA abundance? Maybe unspliced since "they remove the effect of mRNA stability"?
16. Also, for Figure 4C, what do "input" and "computation" mean here?

17. For the discussion of G0 cells in section "characterization of cycling cells shifting to the cycle-arrested state", shouldn't G0 be outside the cell cycle? I'm a little bit confused why these hypothesized G0 cells are mapped as within the mid-G1 phase. Could the authors please comment? I'm particularly concerned about how the G0 cells will be treated. Please also see my comment to the discussion section below.

18. For the discussion section, I'm wondering how DeepCycle will handle cells that are at G0 stage, e.g. terminally differentiated PBMCs. A further question would be, considering an ensemble of cycling and non-cycling cells, can DeepCycle distinguish them? The reason I'm asking is sometimes cycling cells might form separate clusters and cause artifacts for cell type analysis. I totally understand if DeepCycle has difficulty in handling G0 cells of the same/different types, since 1) the current design only takes cell cycle genes, and 2) the latent space only has one variable. Maybe the authors should only focus on cells within the cell cycle by adding a cellular state filter prior to DeepCycle analysis? This will make the study more focused without losing the novelty of the algorithm. Could the authors please comment?

Reviewer #4:

Remarks to the Author:

Riba et al describes DeepCycle, based largely on RNA velocity concept, to infer the transcriptional phase (θ) in relation to the cell cycle based on scRNA-seq data obtained from mESCs, ductal cells and human fibroblasts. This is a method that supposed to be presenting "to the scientific community a broader understanding of RNA velocity and cell cycle maps, that we applied to pluripotency and differentiation". The idea of transcriptional phase (θ) in relationship to cell cycle is interesting, but the execution of the whole manuscript is poor -- it is very difficult to understand what was done leading to the conclusion of many important points as detailed below. A big problem in the experimental design is the comparison between the three chosen cell types that are vastly different (e.g. Line 270-271 "From a general perspective, a clear pattern emerges by comparing the undifferentiated mESCs with the more differentiated human fibroblasts and ductal cells."). While I understand that training and actual application of the model is desired to be done in diverse cell types, there is no confidence that much of the comparisons, therefore the conclusions, are valid, especially in the absence of any biological validation or testing in datasets that have been reported/validated by others.

Specific points:

1. Line 89 indicates that the sequencing depth is uncommon in most scRNA-seq datasets. How deep does sequencing need to be done for DeepCycle to be applicable? This is a relevant and important question that the authors did not address, which questions the general utility of their method. In the abstract (line 22), the authors claim that they "can observe cycling patterns in the unspliced-spliced RNA space for every gene." This statement is misleading and/or overclaiming, as line 91-92 spells out that only several thousand genes can be detected in each cell types, as expected from typical scRNAseq. Further, how would lowly expressed genes compare in performance as highly abundant genes? Gene expression level seems to matter as it was pointed out later that low expression level seems to create inconsistency (line 212-214).
2. Line 97-98 "Cycling genes are expected to be characterized by fully circular patterns as they complete both their activation and deactivation phases (Figure 2A)." and line 116-117 "We expect that genes whose expression is regulated during the cell cycle show a closed path in the unspliced-spliced RNA space consisting of both an active and inactive phase". What would a non cell cycle regulated gene look like? A negative control would be informative.
3. Line 111-112: Figure S2 is used as support that "the complexity of gene regulation in the context of the cell cycle cannot be approximated by the current models". There is hardly any explanation or quantitative measurement for the poor performance of the existing models: how much deviation (Fig S2A) and how much inconsistency (Fig S2B) is driving such a conclusion? Similarly, Cyclum is dismissed without sufficient explanation – very little description is provided for Fig S6 provided.
4. Fig 2E: Why would different cell types show Ccn E and B at different θ ? Are the authors suggesting that these classical cell cycle drivers function differently across cell types? Similarly for the claim later at line 208-210 "other DNA replication genes, such as components of the Origin

recognition complex proteins (Orc1-6/ORC1-6), show different expression patterns across the datasets, suggesting more heterogeneous regulation (Supplementary Figure S10).” These seem to be rather unusual insights that need further elaboration and validation, absence of which question their validity.

5. Line 160-161: No explanation provided for Fig S7 – what criteria is used to conclude “the cell cycle scores calculated by scVelo match well with the transcriptional phases inferred by DeepCycle”?

6. Line 175-176: “the variability of the transcriptomes across the transcriptional phase is stable (Figure 3C)” – this conclusion is based on what? How was “transcriptome variabilities (χ^2)” calculated? Line 178-179 “it could infer the correct dynamics of transcriptional changes at the cell level (see the velocity plots in Supplementary Figure S8)” – what is the analysis that led to this conclusion?

7. Line 183-185 Fig S9 is being used as evidence to support that DeepCycle performed to correctly call the correct proportion of cells in each cell cycle phase. The evidence in Fig S9 is correlation at best. To directly test the performance of DeepCycle, cells should be sorted according to known live cell cycle phase reporters and perform scRNA-seq. These data then should be used to test how well DeepCycle is working. Same problem with Fig 5B,C.

8. Fig 4C: Are the authors suggesting MEK/ERK to be active in pluripotent stem cells (green box)? It is commonly accepted that the inhibition of MEK/ERK that maintains pluripotency, which is also the condition that seems to have been used by the authors, i.e. the “2i” condition. This seriously calls into question of the validity of the results.

We thank the reviewers for their positive and constructive feedback. We have answered their comments in a deeply revised version of the manuscript (see point-by-point discussion below). Among the most prominent changes are:

1. The rewriting of several parts of the paper to gain clarity. In particular, we think we were able to show in a more convincing manner how DeepCycle outperforms other existing algorithms.
2. The development of an automatic approach to detect the cell cycle transitions between phases.
3. The experimental validation of the gene expression patterns predicted by DeepCycle in different cell-cycle phases by bulk RNA-seq experiments on FACS-sorted mESCs.

We thank again the reviewers since we believe the manuscript is now strongly improved as a result of this revision.

REVIEWER COMMENTS

Reviewer #1 (Expertise: DL for scRNASeq data analysis):

In this manuscript, the authors generated in-house single-cell RNA-seq data in mouse ESC and human fibroblast in order to characterize cell cycle dynamic. To analyze the data, the authors propose using an autoencoder approach called DeepCycle to compress the high-dimensional input (spliced UMI + unspliced UMI per gene for 2N input size) into a one-dimension manifold theta.

Strength of the paper:

- Overall, the paper is well written and easy to follow for general audience
- Authors have done extensively exploratory analysis by sorting cells based on theta and observe cells are mainly sorted by their cell cycle phase, which is biologically interesting.
- All figures are of high quality and nicely displayed with clear labels

Main comments:

1. The authors should tone down the language of novel method because the method is essentially a straight application of autoencoder, and no obvious methodological innovation is made.

In the new version of the manuscript, we ensured to clarify the structure of the neural network and the procedure followed to obtain the results showing the important differences with a just straight application of a simple autoencoder. It is important to notice that the special structure and the two training steps adopted are crucial to fit the cycling patterns observed in the unspliced-spliced RNA space and to obtain a meaningful cell-cycle transcriptional phase for each cell. This important point and the ablation analysis in the next answer make us believe that it is possible to claim methodological innovation (see the new Methods 'Implementation of DeepCycle' and the Supplementary Figure S3).

2. If authors do want to claim method novelty, quantitative evaluation such as ablation analysis is needed. In particular, how does DeepCycle perform when only spliced or unspliced read (but not both) are provided in terms of recapitulating known cell phase cycle?

As suggested by the reviewer, we performed an ablation analysis showing that using only unspliced or spliced generates inconsistent results with the other cell cycle measures (e.g. cyclin-E and -B expressions, RNA counts per cell) and the estimation of the transcriptional phase (see new Supplementary Figure S7). Interestingly, using only the unspliced reads return a reversed transcriptional phase that could lead to wrong order of expression of cell-cycle genes. Moreover, it is not possible to establish if the fitting procedure for the two ablations worked properly since the gene-specific cycling patterns are only apparent in the joint unspliced-spliced RNA space. One of the advantages and methodological novelties of DeepCycle is exactly to exploit the cycling patterns to obtain a biologically meaningful fitting and have a validation on the fitting procedure that can be supervised by the user.

3. Only one method called Cyclum is compared with DeepCycle and showed poor UMAP clustering (Suppl. Fig. 6). How does DeepCycle (which is just an autoencoder) compare with other approaches such as scVI, scVI-LD, scGAN, scETM, which properly takes into account distribution of the latent

encoded variable in a variational autoencoder framework while addressing batch effects in different ways.

Regarding the mentioned methods, DeepCycle is not an imputation tool and works explicitly only to detect the cell cycle. We clarified this in the main text, so it is not comparable to scVI, scVI-LD, scGAN and scETM. In the main text, we added a discussion about the imputation methods where we envision a possible extension of DeepCycle to a Variational AutoEncoder. Thanks to VAE, DeepCycle could take into account the distribution of the latent encoded variable (e.g. FACS distribution) but as we performed one first attempt to implement DeepCycle as a VAE, the neural network was not converging to the correct result.

Supplementary Figure S6 shows the 2d projection of the angle (transcriptional phase) inferred by Cyclum, DeepCycle, and random uniform distribution of angles. For clarity, we have added an example of what one should expect when Cyclum correctly identifies the cycling signature of the data.

4. How was the autoencoder network architecture derived? Was a validation set used here in terms of reconstruction loss?

DeepCycle is mainly composed of Dense layers (fully connected layers) that are the most basic structures for a neural network. There are some important additions: 1) the circularization layer to transform the output of the encoder into a circle; 2) the estimated phase from the input gene was included as an additional input to the last Dense layer of the encoder; 3) the Gaussian noise layers to avoid overfitting.

Yes, 17% of the input cells are used as validation.

We now significantly expanded the Methods and Supplementary Materials to provide extensive detailed information on the autoencoder structure and the training procedure.

5. Given known gene markers, can authors show the model can predict accurately the cell cycle phase in a classification tasks using standard quantitative metric such as prediction accuracy? This may help others to annotate their single-cell data with unknown cell cycle phases.

Since the lack of labels, we could not train any model to predict accuracy. Instead, we focused on developing a strategy to automatically detect the cell cycle transitions. The detection uses the transcriptional phase and cell cycle gene markers, see the new Supplementary Figure S9 and Methods section 'Detection of the cell cycle phase transitions'. It has also been implemented in python and distributed in the Github repository.

Reference:

1. Lopez, R., Regier, J., Cole, M. B., Jordan, M. I. & Yosef, N. Deep generative modeling for single-cell transcriptomics. *Nature Methods* 15, 1–11 (2018).
2. Svensson, V., Gayoso, A., Yosef, N. & Pachter, L. Interpretable factor models of single-cell RNA-seq via variational autoencoders. *Bioinformatics (Oxford, England)* 36, 3418–3421 (2020).
3. Bahrami, M. et al. Deep feature extraction of single-cell transcriptomes by generative adversarial network. 570, 1–21 (2020).
4. Zhao, Y., Cai, H., Zhang, Z., Tang, J. & Li, Y. Learning interpretable cellular and gene signature embeddings from single-cell transcriptomic data. *bioRxiv* 2021.01.13.426593 (2021).

Reviewer #2 (Expertise: deep learning, biomedical data, single cell data):

The manuscript 'Cell cycle gene regulation dynamics revealed by RNA velocity and deep-learning' describes a deep learning-based algorithm (Deep Cycle) to predict cell cycle state from scRNA-seq velocity data. The algorithm can be used to analyze genes for their cell cycle involvement once fit, including TFs and potentially novel cycle-regulated genes.

While not being a cell cycle specialist, I think the validation experiments and biological insights gained with Deep Cycle are compelling. The authors provide good evidence that their algorithm outperforms other existing tools in cell cycle fitting. The paper is written in good English.

While I think the biological part is convincing (while not being an expert) I think some other parts could improve or have shortcomings.

Major points

1) I think the technical sections are way too short to be understandable. In other words, I think the description of the model and training of it are insufficient. Layer sizes, primary training, why two training steps? How do you map from vectors to an angle, do you make sure to keep unit length vectors through layer norm? What optimizer, loss function, when do you stop training? None of this information is supplied in the git page, one would have to dig deeply into the code to figure everything out.

The Methods section about DeepCycle has been extended providing all the necessary details and a more comprehensive Supplementary Figure S3 shows the fine structure of the autoencoder and the training parameters. Also, the GitHub repository has been updated and contains detailed information about the structure of the network and the training procedure.

- The sizes of the layers are specified in Supplementary Figure S3 and are equal to 4 times the number of genes.
- The first step is used to provide an initial guess of the transcriptional phase to the encoder and trains the encoder to predict a phase from the provided input gene. The second step trains both encoder and decoder with a standard autoencoder training.
- We didn't use any layer normalization since the domain of an angle is $(-\infty, +\infty)$ and it can always be mapped back to $2\pi * [0, 1]$.
- The mean squared error was used as a loss function and minimized using Adam, which was the best performing optimizer among the ones tested (SGD, RMSprop, and Adam). In addition, an early stopping criterion was used when the loss function evaluated on the validation data reached a plateau (see details in Methods and Supplementary Figure S3.).

2) When I read up on Cyclum (to understand how different their algorithm might be to Deep Cycle) I came to understand that most of Deep Cyclum seems to be based on Cyclum. Again, this is a bit tricky to judge as there is hardly any information on the Deep Cycle algorithm but suffice it to say that I started to understand how Deep Cycle might work by reading the Cyclum paper. The main difference seems to be that Deep Cycle uses RNA velocity information as input from scVelo, Cyclum does not. In other words, DeepCycle's algorithmic novelty seems incremental, at best. It bothered me that the authors don't clearly state that their 'algorithm' is heavily based on Cyclum, while I have to admit that it is hard to judge, due to the extremely limited technical information given in this paper.

DeepCycle uses the unspliced-spliced values and does not use the velocities as inputs. We added in the revision the ablation analysis, see Supplementary Figure S7, which shows that using only the spliced or unspliced gives inconsistent results and you're not assured to have the right cell cycle directionality. Further, each run of Cyclum returns different results without any warnings, so not only the algorithm has several pitfalls but also the implementation is not stable and it cannot be controlled (Supplementary Figure S6). On the other hand, combining unspliced-spliced information allows DeepCycle to infer the right directionality and can be supervised by checking the cycling genes. Thus, the novelty of DeepCycle relies on the combination of theoretical expectations with data leading to the estimation of a transcriptional cell-cycle phase that is biologically meaningful as opposed to returning incoherent and inconsistent results. Importantly, the DeepCycle results can easily be scrutinized and the user knows immediately when to trust the estimated cell cycle phase.

Minor points

1) To me, it is really hard to understand what exactly this paper is about purely from the abstract. It is very high-level claims and no examples or concrete results. I would definitely rewrite this a bit.

The abstract has been rewritten almost completely to state the aim of the paper and the main results. In addition, we have included a concise but intuitive explanation of the key rationale behind DeepCycle.

2) Many figures have very short legends, to the degree that it is virtually impossible to understand what is shown in the figures by just reading the legends. Examples are: Figure S3 - no details on the three inputs, no details on the sizes of the layers (no activation information), no info on the sigma layer; Figure S6 - no mention of A and B, no description of what x and y are, no mention of which data was used. These are just two examples but this lack of description is pervasive.

We thank the reviewer for pointing out the lack of clarity and in the revised version of the manuscript, we extend all the Supplementary Figures and the related captions.

3) The introduction is incoherent and not complete. The authors talk early on about the 'unspliced-spliced RNA space' without ever mentioning what this is and how this is relevant to cell cycle prediction. They never mention the RNA velocity theory in the introduction, which is the basis of this paper. When they mention the theory in the results section, they don't explain what it is (I had to piece it together from the text).

The introduction has been rephrased to accommodate a sentence about the RNA velocity idea and link it to the cell cycle and the cycling genes (red text).

Reviewer #3 (Expertise: DL for scRNASeq data analysis):

Riba et al. proposed DeepCycle, an autoencoder model which uses a single latent variable to describe the cell cycle status. DeepCycle is conceptually novel and might potentially provide great biological insights. The DeepCycle algorithm per se was well described and the corresponding benchmarking analysis was rigorous. However, I think further clarification is needed for the biological applications, especially in sections "identification of cell-cycle core transcription factors" and "characterization of cycling cells shifting to the cycle-arrested state", to make the study scientifically sound. I thus have the following comments:

1. Throughout the paper, the dot size in scatterplots might be too big. I fully acknowledge that the authors used density plots to describe the distribution of cells, but it would be better if the authors could also provide a clear representation of individual cells.

The size of the dots in the main scatter plots presenting the datasets, has been decreased to better highlight the number of cells.

2. For Figure 1D, the scrutiny of data quality is highly appreciated. Based on my experience, in general 1) primary cells have less mRNA compared to cultured cells, 2) human cells have more mRNA compared to mouse cells, and 3) "stem-like" cells have more mRNA compared to "differentiated" cells. So the #UMIs per cell differences might be a piece of "real biology", rather than only a quality issue. Since they authors also provided spliced/unspliced% reads in Figure 1E so I'm convinced about the quality of the datasets used here.

Thanks

3. For Figure 1F, are the two axes represent the fraction of spliced/unspliced reads? Or some integrated "velocity scores" from the spliced-unspliced space? A short explanation would be really appreciated. The same comment applies to other "velocity plots".

The two axes represent the z-score values for unspliced and spliced reads. The z-score values are necessary to input to DeepCycle values that are around 0 and with a similar range across different genes. This information has been added to the plots in the main figures.

4. For citations 21-28, considering DeepCycle is based on AE, I would suggest the authors only cite other algorithms based on AE, since VAE is generative model and might not be directly comparable.

We discussed in the revised version the possibility to extend DeepCycle to a VAE and make sure to clarify the difference.

5. For Figure 2C, The "un" represents abundance of unspliced gene #n, and the "sn" represents abundance of spliced gene #n, correct? I think the authors should clarify the two terms in the figure caption.

Thanks for noticing the missing information, this has been clarified in the corresponding caption.

6. For Figure 2C, the red circle was constructed from the theta angle latent variable, correct? Then my question is how the authors correspond sin/cos theta trajectory on the "velocity space".

We reconstructed the red circular pattern by mapping the latent variable (angle/theta) with the decoder to the spliced-unspliced space.

7. For Supplementary Figure S3, the decoder takes u and s to calculate the θ , meanwhile imputes Gaussian noise to the data to increase the robustness, correct? As for the decoder, it recapitulates the expression matrix by “circularization” from θ , correct? I think the authors should provide a more detailed explanation to the framework. Some intuitions on the design strategy of the framework would be highly appreciated.

Yes, one side of the encoder uses an estimation of the phase from the input gene and feeds it to the last Dense layer of the encoder itself (see the orange box in Supplementary Figure S3). This step has been added to provide the neural network a first guess of the correct θ . θ is then circularized as a first step of the decoder to enforce a 1-dimensional circular manifold. We explained more extensively the framework and the training procedure in the Methods and Supplementary Figure S3.

8. For "GOterm:cell_cycle", have the authors tried using all genes? The reason I'm asking is that the scLVM paper (<https://www.nature.com/articles/nbt.3102>) claimed other genes might also be affected by cell cycle. Further, maybe extending the analysis to all genes will introduce additional biological features besides cell cycle in the latent variable, especially when analyzing multiple cell types together? Could the authors please comment?

We observed a trade-off between signal and noise. A restricted subset of genes that are truly cycling makes the fit more robust and, afterward, it is possible to identify unknown cycling genes.

9. For Supplementary Figure S4, I think the authors should specify which dataset is presented in the figure.

11. For Supplementary Figure S6, again, I think the authors should specify which dataset is presented in the figure.

In both analyses, the mESC dataset was used. We explicitly write it in the figure captions.

10. For Supplementary Figure S5, I would suggest the authors to describe how the “cell cycle score” represented by the x-axis is calculated. Also, I would suggest the authors to use density plot here, considering there is an aggregation of dots at the top-left corner. The authors should confirm that most of the dots are on the diagonal.

Supplementary Figure S5 has been updated with histograms to show that the distribution of transcriptional phases is rather uniform on both axes. The ‘accumulation’ of dots in the corners is due to the fact that the estimated θ is shifted by an arbitrary angle (in this case small).

After each run of DeepCycle, the estimated phase might be shifted, so to clarify this we added a plot representing the possible expected behaviors in case of robust phase inference and extended its caption.

12. For Supplementary Figure S7, are the authors using the same UMAP layouts as Figure 3A and Supplementary Figure S8? Seems the same layouts are used so the claim here is valid. I would suggest the authors to put the UMAP plots color-coded by scVelo scores and DeepCycle θ values side-by-side to better support the claim here.

We joined together the figures into Supplementary Figure S10 and a third column has been added showing the S and G2M scores from scVelo across the transcriptional phase.

13. For the flow cytometry analysis, I really appreciate the experimental validation here. However, it seems that the authors didn't describe the cell staining strategy. Was Hoechst staining performed to determine cell cycle stages?

We performed propidium iodide staining as reported in the Methods. The information has been added in the main text, accordingly.

14. For section "identification of cell-cycle core transcription factors" in general, I think the correct way of demonstrating “identification” would be 1) identify top TFs that are correlated with θ , and 2) show these top TFs are biological meaningful. So I think the authors should either 1) perform a real “identification” analysis, or 2) only claim they could recapitulate known biology as an additional support for validating the θ value.

We rephrased the section title as 'Prediction of cell-cycle core transcription factors'. For the rest of the sections, we make sure not to overstate the results of the analysis.

15. Specifically, for Figure 4B, are the r values calculated against the abundance of all, spliced or unspliced mRNA abundance? Maybe unspliced since "they remove the effect of mRNA stability"?

The r values are computed between the motif activity and the spliced RNAs since we expected the latter to reflect better the level of stable proteins, compared to the unspliced RNAs. The caption has been extended accordingly.

16. Also, for Figure 4C, what do "input" and "computation" mean here?

Dunn et al Science 2016 identified the transcriptional network responsible for the maintenance of the pluripotent state in mESC. The TFs identified as input are responsible to integrate the pluripotency maintenance signal and the 'computation' ones to make a decision about the cell fate. We removed the panel of 'input' and 'computation' because it generated some confusion and clarified this in the caption.

17. For the discussion of G0 cells in section "characterization of cycling cells shifting to the cycle-arrested state", shouldn't G0 be outside the cell cycle? I'm a little bit confused why these hypothesized G0 cells are mapped as within the mid-G1 phase. Could the authors please comment? I'm particularly concerned about how the G0 cells will be treated. Please also see my comment to the discussion section below.

Notice that in the case of human fibroblasts two subpopulations (proliferative and non-proliferative) were clearly observed from the beginning. Thus, we trained DeepCycle using only proliferative cells, and cells in G0 were excluded. Afterward, we run the trained DeepCycle on G0 cells as if they were proliferating cells to assess what phase DeepCycle would assign. Interestingly, they were mapped to the early G1 phase as one would expect due to the fact the cell cycle tends to stop in G1. One possible extension of DeepCycle that we envision in the future would be to automatically classify proliferative and non-proliferative cells in a first step and then estimate the transcriptional cell-cycle transcription phase of cells from the proliferative subpopulation.

18. For the discussion section, I'm wondering how DeepCycle will handle cells that are at G0 stage, e.g. terminally differentiated PBMCs. A further question would be, considering an ensemble of cycling and non-cycling cells, can DeepCycle distinguish them? The reason I'm asking is sometimes cycling cells might form separate clusters and cause artifacts for cell type analysis. I totally understand if DeepCycle has difficulty in handling G0 cells of the same/different types, since 1) the current design only takes cell cycle genes, and 2) the latent space only has one variable. Maybe the authors should only focus on cells within the cell cycle by adding a cellular state filter prior to DeepCycle analysis? This will make the study more focused without losing the novelty of the algorithm. Could the authors please comment?

Currently, DeepCycle is designed to work on cycling cells, if the population of cells is too heterogeneous the overlap of processes different from the cell cycle will mask the cell cycle signature. We extended the analysis to automatically detect the transitions between cell phases, see the section in the Methods, 'Detection of the cell cycle phase transitions'. We are exploring the possibility to infer the proliferative state from marker genes but did not find a good general measure yet.

Reviewer #4 (Expertise: Stem cell biology):

Riba et al describes DeepCycle, based largely on RNA velocity concept, to infer the transcriptional phase (θ) in relation to the cell cycle based on scRNA-seq data obtained from mESCs, ductal cells and human fibroblasts. This is a method that supposed to be presenting "to the scientific community a broader understanding of RNA velocity and cell cycle maps, that we applied to pluripotency and differentiation". The idea of transcriptional phase (θ) in relationship to cell cycle is interesting, but the execution of the whole manuscript is poor -- it is very difficult to understand what was done leading to the conclusion of many important points as detailed below. A big problem in the experimental design is the comparison between the three chosen cell types that are vastly different (e.g. Line 270-271 "From a general perspective, a clear pattern emerges by comparing the undifferentiated mESCs with the more differentiated human fibroblasts and ductal cells."). While I understand that training and actual application of the model is desired to be done in diverse cell types, there is no confidence that much of

the comparisons, therefore the conclusions, are valid, especially in the absence of any biological validation or testing in datasets that have been reported/validated by others.

We indeed believe that showing that DeepCycle works in vastly different cell types is compelling evidence of the applicability of our method. Notice that the fact that our approach doesn't require synchronizing or genetically modifying cells, makes it suitable to study gene regulation during the cell cycle *in vivo*, from model organisms to even patients. In this new version of our manuscript, we have rewritten several parts of the paper to gain clarity and in particular we have shown in a more convincing manner how DeepCycle outperforms other existing algorithms. More importantly, we have experimentally validated the gene expression patterns predicted by DeepCycle in different cell-cycle phases by bulk RNA-seq experiments on FACS-sorted mESCs. All together, we strongly believe that DeepCycle is able to produce reliable gene expression dynamics throughout the cell cycle in unperturbed cell populations which may open the possibility to study cell-cycle related gene regulation in a myriad of systems in the future.

Specific points:

1. Line 89 indicates that the sequencing depth is uncommon in most scRNA-seq datasets. How deep does sequencing need to be done for DeepCycle to be applicable? This is a relevant and important question that the authors did not address, which questions the general utility of their method. In the abstract (line 22), the authors claims that they "can observe cycling patterns in the unspliced-spliced RNA space for every gene." This statement is misleading and/or overclaiming, as line 91-92 spells out that only several thousand genes can be detected in each cell types, as expected from typical scRNAseq. Further, how would lowly expressed genes compare in performance as highly abundant genes? Gene expression level seems to matter as it was pointed out later that low expression level seems to create inconsistency (line 212-214).

The majority of the scRNA-seq datasets we analyzed do not show genes with cycling patterns, for either technical (shallow sequencing), biological reasons (lower number of expressed mRNAs, not proliferating cells, etc), or both. The mESC and the human fibroblasts are less differentiated cell types compared to ductal cell progenitors and, this reflects in a large amount of expressed mRNAs as we could infer from the low saturation levels (~20%) from the 10x Chromium platform. Our current recommendation is to have numbers comparable to our mESC and human fibroblast datasets, but even with datasets as shallow as the ductal cell, the method can give reliable results (~3k genes and ~8k UMI per cell).

The inclusion of genes with low expressions and therefore noisier does not allow the neural network to converge. For that reason, we implemented a filtering step (flag --hotelling) to select among the list of cell cycle genes the ones showing higher variability in expression (see Methods 'Identification of cycling genes and high-density paths'). We rewrote most of the abstract to avoid misunderstandings and gain clarity.

2. Line 97-98 "Cycling genes are expected to be characterized by fully circular patterns as they complete both their activation and deactivation phases (Figure 2A)." and line 116-117 "We expect that genes whose expression is regulated during the cell cycle show a closed path in the unspliced-spliced RNA space consisting of both an active and inactive phase". What would a non cell cycle regulated gene look like? A negative control would be informative.

By selecting only genes whose spliced expression is above 1, we can detect that the majority of the genes in the 3 datasets are quite stable in expression levels across the cell cycle. Among these genes the genes showing 2-fold change across the cell cycle are 218 out of 790 for the ductal cells, 442 out of 3533 for the fibroblasts, and 116 out of 4101 for the mESC. This information has been included in the caption of Supplementary Figure S15. We added examples of non-cycling genes in Supplementary Figure S8.

3. Line 111-112: Figure S2 is used as support that "the complexity of gene regulation in the context of the cell cycle cannot be approximated by the current models". There is hardly any explanation or quantitative measurement for the poor performance of the existing models: how much deviation (Fig S2A) and how much inconsistency (Fig S2B) is driving such a conclusion? Similarly, Cyclum is dismissed without sufficient explanation – very little description is provided for Fig S6 provided.

We added in Figure S2 the expected behavior of the models when identifying the correct cell-cycle dynamics. ScVelo captures neither the circular patterns nor the correct latent time (bold purple lines in

Figure S2). Regarding Cyclum Figure S6, we added the results expected from a random distribution of angles and DeepCycle to clarify that Cyclum does not work properly. Further, running Cyclum multiple times returns different angle distributions that suggest it is unstable and unreliable.

4. Fig 2E: Why would different cell types show Ccn E and B at different θ ? Are the authors suggesting that these classical cell cycle drivers function differently across cell types? Similarly for the claim later at line 208-210 “other DNA replication genes, such as components of the Origin recognition complex proteins (Orc1-6/ORC1-6), show different expression patterns across the datasets, suggesting more heterogeneous regulation (Supplementary Figure S10).” These seem to be rather unusual insights that need further elaboration and validation, absence of which question their validity.

Theta represents an arbitrary state across the cell cycle and is not shared between datasets. The full cycle in the theta space has a length of 1 for all the cell types. We synchronized the thetas to have mitosis in 1, see the section in the Methods, ‘Detection of the cell cycle phase transitions’. Now, if in real-time the length of the cell cycle is different we should see mainly a change in the G1 phase. According to the literature, the S and M phases are quite constant, constrained by the structural events happening in the cells, and do not depend on the different cell types. To have a more realistic visualization of the length of the cell cycles we set the same length for the S phase and rescale accordingly the thetas (see Supplementary Figure S12). It becomes clear that the cell cycle lasts much longer in ductal cells compared to mESCs and that CcnE and B show similar dynamics with respect to the G1/S and M/G1 transitions respectively.

We couldn’t provide more evidence about the Origin Recognition Complex so decided to remove it from the text.

5. Line 160-161: No explanation provided for Fig S7 – what criteria is used to conclude “the cell cycle scores calculated by scVelo match well with the transcriptional phases inferred by DeepCycle”?

We merged Supplementary Figures S7 and S8 into S10 and added a comparison between the scores and the transcriptional phase to improve the clarity of the statement. The Supplementary Figure S10 now includes a plot of the S and G2M scores, inferred by scVelo, across the transcriptional phases for the three datasets.

6. Line 175-176: “the variability of the transcriptomes across the transcriptional phase is stable (Figure 3C)” – this conclusion is based on what? How was “transcriptome variabilities (χ^2)” calculated? Line 178-179 “it could infer the correct dynamics of transcriptional changes at the cell level (see the velocity plots in Supplementary Figure S8)” – what is the analysis that led to this conclusion?

The chi-squared has been calculated using the expected value of the average for the bin. Anyway, In the last version of the draft, we decided to remove panel 3C to improve the flow of the text since it was not adding much information to the manuscript. The velocity maps are consistently pointing in the direction in which the transcriptional phase is increasing (see Supplementary Figure S10). For that reason, we can conclude that at the cell level the velocity estimation is consistent with our estimation of the cell cycle progression.

7. Line 183-185 Fig S9 is being used as evidence to support that DeepCycle performed to correctly call the correct proportion of cells in each cell cycle phase. The evidence in Fig S9 is correlation at best. To directly test the performance of DeepCycle, cells should be sorted according to known live cell cycle phase reporters and perform scRNA-seq. These data then should be used to test how well DeepCycle is working. Same problem with Fig 5B,C.

We agree with the reviewer that a nice validation would be to sort live cells according to phase reporters, like FUCCI, but we would need to repeat the experiment with a cell line that is not comparable with our dataset since the introduction of reporters. As an alternative, we bulk sequenced mESCs sorted with FACS in the three main phases, showing the results are consistent with our cell cycle phases in Supplementary Figure S11. Further, we extensively checked cell cycle genes (Fig. 2E, 3A Supplementary Figure S14), general cell cycle markers (Supplementary Figure S10), and the cell sizes (Figure 3C). All these together strongly indicate, we believe, that DeepCycle is working properly successfully sorting cells according to their cell-cycle progression.

8. Fig 4C: Are the authors suggesting MEK/ERK to be active in pluripotent stem cells (green box)? It is commonly accepted that the inhibition of MEK/ERK that maintains pluripotency, which is also the

condition that seems to have been used by the authors, i.e. the "2i" condition. This seriously calls into question of the validity of the results.

The genes reported in the boxes are from <https://science.sciencemag.org/content/344/6188/1156> and correspond to the transcription factors relevant for the maintenance of pluripotency. We used this list to select relevant TFs and observe their activity across the cell cycle Figure 4C. Unfortunately, the motif of MEK/ERK is not in the database we used to infer the TF activities and it is indeed missing from the heatmap 4C. The main idea was to show the subsequent activation of the input genes during the late G2M phase and the computation genes in early G1. To avoid any confusion, we decided to remove the list of TFs.

Reviewers' Comments:

Reviewer #1:

Remarks to the Author:

1. A general comment on the style of the response letter. It would be a lot more helpful to write down the revised content as part of the response as opposed to directing the reviewer to general section.
2. VAEs are not "imputation methods to correct for capture rate and noise in scRNA-seq data". They are general Bayesian framework to infer the posterior distribution of the latent manifold from typically high-dimensional data [Kingma & Welling, 2013]. The authors wrote in the main text under Results section that they tried VAE but not able to obtain a convergent neural network without showing any results. I found this very unsatisfactory especially that there is no detail provided in how the VAE was implemented and what defines "convergent" in their VAE model. In VAE, the objective function is evidence lower bound that is the reconstruction loss (i.e., likelihood) minus Kullback-Leibler divergence (i.e., $KL[q||p]$). What exactly do you mean by "We tested VAE but without being able to obtain a convergent neural network"?
3. Authors mentioned that the lack of gold-standard labels for cell-cycle does not allow them to do proper method comparison. But in Figure 3A,C and Fig 4A they did label the cells with cell cycles based on marker genes. What I was asking in original comment was to define cells based on the marker genes and then the Theta to predict these cells.
4. On line 463,464, the symbols do not show up properly on the pdf file probably due to conversion issue.

Reviewer #2:

Remarks to the Author:

The authors have done a good job addressing the comments of the reviewers. The algorithmic details have sufficient depth in writing and in the figures. The github page and abstract are heavily extended / modified. Overall I think this work could be published in its current state, with minor fixes of spelling and grammatical mistakes. I have no further questions or concerns.
Stefan Bonn

Reviewer #3:

Remarks to the Author:

All my concerns have been addressed. Besides, the authors also did great amount of additional work to significantly improve the quality of the paper, which I really appreciate. I thus would recommend the publication of the paper.

Two minor comments:

1. It seems that the theta cannot be displayed, e.g. line 123.
2. The genes listed at the end of the main text file should be Supplementary Table S1 "cycling genes not yet considered in the GO term:cell_cycle that could be added as markers of the cell cycle", right?

Reviewer #5:

Remarks to the Author:

Review of Ms entitled "Cell cycle gene regulation dynamics revealed by RNA velocity and deep-learning" by Riba et al submitted to Nature Communications^[1]_[SEP]

Summary:

In this study, the authors use a computational analysis of deep scRNAseq data to assign the cell cycle phase of individual cells in unperturbed populations. Using RNA velocity, and parameters

derived from the ratio of spliced to un-spliced transcripts they have projected the dynamic temporal activity of individual loci in single cells. The main finding is that the temporal oscillations of cell cycle-regulated genes can be detected by projecting a trajectory from (static) single time-point data of a cycling population to yield a specific signature of cyclic transcriptional activity in the "RNA velocity space". This index can be used to infer the cell cycle phase of individual cells. The study is performed in mouse embryonic stem cells (mESC) and somatic human fibroblasts (IMR90), and then tested using published data from ductal progenitor cells.

The study provides a potentially very useful tool for deriving cell cycle status from single cell transcriptomic data. Overall, the results are intriguing and the authors make some compelling arguments for the utility of this approach. However, there are a number of issues that require clarification, and potentially would make the study more accessible and therefore more widely applicable.

I will restrict my comments to the biology of the systems used, and the biological inferences drawn from the computational approaches. Overall, I found that the paper is written more for the computational biologist than the cell biologist. From the viewpoint of the biologists who might find the approach very useful, and in the interest of wider applicability, there are several statements/inferences/conclusions that would benefit from a more detailed explanation.

Example 1: The statement "The fits naturally generate gene expression series that can be analyzed to obtain detailed kinetic parameters" ^[1]_{SEP}

An explicit statement is essential to explain how dynamic patterns can be computed from static single time-point data to infer cyclic activity, else it is difficult to appreciate the results.

Example 2: There is a very basic explanation of what TFs do, which I presume is for the computational expert, but the equivalent basic treatment of very complex computational inferences for the cell biologist are completely missing.

The Introduction does not provide the uninitiated with sufficient insight as to what 'RNA space' may refer to, why un-spliced vs. spliced reads are more useful than total reads, and how RNA velocity theory has come to be an accepted mode of investigation of transcriptomic data. Some explanation is found in the Discussion but in my opinion, this is better positioned in the Introduction so as to provide a clear framework from which to view this study. Can be reiterated in the Discussion if space is not an issue.

From a biological perspective, it is not clear that the choice of three very different cell types is relevant. It would have been better to use comparisons between different states of a single cell type (for example undifferentiated mESC with their differentiated derivatives robust protocols exist for different lineages such as cardiac or neural or endothelial). From such data if the cell cycle changes that accompany differentiation (and cell cycle) could be mapped using DeepCycle, it would have been more convincing, and if then applied to mouse or human somatic cells such as fibroblasts or other cell types, one could appreciate the broader generalizability of the method. However, given that substantial work has been done with the 3 very diverse cell types compared in the current ms., it would be important to tone down the claims of biological relevance and generalizability and restrict them to the specific results obtained.

A major concern is the derivation of inferences about the behavior of G0 (quiescent) cells from a data set derived from a continuously cycling population.

The authors make a strong point about the application of their method to cells that have not been perturbed (synchronized) with drugs or by engineering with fluorescent markers. However, designation of a sub-population as G0/G1 based on FACS refers to the 2n DNA content (even a DNA/RNA ratio would have been more helpful to grossly distinguish G0 from G1). Data from cycling cells does not take into account the differences in behavior of truly quiescent cells (G0) which have entered into a qualitatively different state based on activation of a specific quiescence program coincident with cell cycle withdrawal. In this study, it is not at all clear that individual cells have actually reached a stationary phase, so it is a stretch to say whether cells are quiescent or will start cycling again, since no phenotyping in terms of kinetics of cell cycle have been done. Designation of a quiescent state relies on not just the DNA/RNA content or transcriptional profile, but critically, the kinetics and expression profile of cell cycle re-entry (the G0-G1 transition) which

distinguishes these cells from G1 cells (even pre-R point), for discussions see Collier 2006; Goodell, 2004; 2006; Dhawan and Laxman 2015).

Therefore, the rather strong statements about whether cells are entering G0 are not justified, and need to be toned down or qualified substantially. G0 markers were derived from the Cheung and Rando data set, but these are from quiescent muscle stem cells (or very early after activation), not a continuously cycling population. It is not clear why the data from human fibroblasts was not used (Collier 2006) or from the core quiescence signature (Qsig) as derived by the Goodell group.

It is important to note that unperturbed populations of quiescent and exponentially cycling cells of the same type can be derived from normal and regenerating adult muscle (see Machado et al, 2017; van Velthoven et al, 2017), and might provide a much more stringent test of the ability of this computational approach to identify cells in different cell cycle phases, particularly G0.

From this perspective, several statements relating to Figure 5 therefore did not seem warranted or were overly emphatic:

Specifically:

(i) FoxM1 being designated as a marker of quiescence when it is specifically down-regulated after M phase is rather strange; normally one would designate an up-regulated gene to be a bona fide marker of a state if used alone; if used in conjunction with a signature, then up and down regulated genes provide more power. Further, since FoxM1 targets do not follow its expression, there may be many layers of regulation, which are not captured by this approach so it is not clear how this statement supports the conclusions.

(ii) One would expect quiescence markers to be highly changed only in truly quiescent cells, not necessarily in the lagging cells of a G1 subpopulation of a cycling population, as their induction comes from extended period of time in G0, activating a new program. So this comparison did not seem compelling.

(iii) While there is a reasonable case to be made for using DeepCycle, many new insights about the quiescence decision point have gained from experiments in the Meyer and Spencer labs, using unperturbed cycling populations of fibroblasts (albeit engineered with fluorescent markers, which I do not believe alters their conclusions). It would be important to acknowledge the utility of that approach.

(iv) Inferring master regulators of G0 from this data feels like quite a stretch- as mentioned above, one cannot really derive strong conclusions about G0 since this is based on G1 cells from a cycling population.

Therefore, the authors may want to substantially tone down their description and inferences from Fig5, and take into account the altered state of G0 cells compared to G1. Indeed, it is important to describe more carefully the entry into quiescence, its maintenance and exit from this type of data. But while some elements of the transition from G1 towards G0 may be captured from these data, I am not convinced that the G0 state is captured.

Can the authors do experiments to show that cells isolated on the basis of the RNA velocity index are kinetically distinct? Where do they fall in the cell cycle?

Some specific issues include:

Is it not surprising that despite differences in read depth and very different transcriptional landscapes, all the data sets show ~15% unspliced reads? Would the ratio not be significantly different at different times/cells? Authors should comment on why regulation at splicing does not appear to be a feature, or provide a reference.

Fig1 # of genes per cell- is it not more correct to express as # of transcriptionally active genes per cell since the gene number per se should be identical for the two mouse cell types and slightly different for human cells?

Fig3D does not exist in the revised version, but has been cited: harmonize

YY1 is both an activator and a repressor-how will that affect inferences?

Fig5 why not use DNA-RNA ratio, which is a better separator of G1 from G0?

Statement: Ybx1 expression is linked to poor prognosis in pancreatic ductal adenocarcinoma –why is this relevant in the context of this paper?

Line 308 ⁽¹⁾_{SEP}“G1 are bigger than the cells in S and G2/M suggesting that the cells waiting in G0/G1 are increasing in size, it remains unclear whether they will re-enter the cycle later. The cell velocities are consistent with our interpretation and do not clarify if the cells in G0 will start cycling again” (Supplementary Figure S16). ⁽¹⁾_{SEP}

Again: designating G1 cells from a cycling population as quiescent is not warranted as they just represent a continuum of lagging G1 not G0

Need explicit statement of why spliced to unspliced transcripts is a better measure of transcriptional status of a given locus than total transcripts

Comparison of ESC and somatic cells is not strictly useful since they have a very different cell cycle structure and regulation and mESC never enter G0

Clarifications required for the following:

Abstract: rather than pluripotent vs. differentiated cells more correct to say embryonic vs. somatic since from the cell cycle perspective both are cycling but with different characteristics

Supp Fig 1: indicate what the blue and orange traces in the gene expression plot (lower panel) represent. The current figure legend has no explanation. This figure is skewed for genes that are highly expressed in cycling cells, where only Dek represents a gene more highly expressed in non-cycling cells- more examples of these should be shown; genes that are known to be expressed only in S phase or early G1 could also be shown to give a sense of the robustness of this approach.

Supp figs often refer to main figs (eg S17), but they should be self-explanatory, and the legends should be more detailed to allow appreciation of the point being made. Perhaps a title based on the inference being drawn from each sup fig can be included to enable understanding?

REVIEWER COMMENTS

Reviewer #1 (Remarks to the Author):

1. A general comment on the style of the response letter. It would be a lot more helpful to write down the revised content as part of the response as opposed to directing the reviewer to general section.

We sincerely apologize for the inconvenience it has caused to the reviewers to not include the revised content in our answers. We thought that highlighting the text in red in the manuscript was enough. This time, to ease the reviewing process we have provided more details directly in the answers below.

2. VAEs are not “imputation methods to correct for capture rate and noise in scRNA-seq data”. They are general Bayesian framework to infer the posterior distribution of the latent manifold from typically high-dimensional data [Kingma & Welling, 2013]. The authors wrote in the main text under Results section that they tried VAE but not able to obtain a convergent neural network without showing any results. I found this very unsatisfactory especially that there is no detail provided in how the VAE was implemented and what defines “convergent” in their VAE model. In VAE, the objective function is evidence lower bound that is the reconstruction loss (i.e., likelihood) minus Kullback-Leibler divergence (i.e., $KL[q||p]$). What exactly do you mean by “We tested VAE but without being able to obtain a convergent neural network”?

We fully agree with the reviewer: in general VAEs are not imputation methods. However, we didn't intend to convey that statement in our manuscript. Below we attempt to clarify this misunderstanding.

The reviewer's original question in the previous revision was: “*How does DeepCycle (which is just an autoencoder) compare with other approaches such as scVI, scVI-LD, scGAN, scETM, which properly takes into account distribution of the latent encoded variable in a variational autoencoder framework while addressing batch effects in different ways*”

To answer this question we wanted first to stress the fact that the VAE methods mentioned by the reviewer are focused mainly on the imputation of missing data and not on inferring the cell cycle progression in single cells. This impeded the direct comparison of the output and performance of these methods with DeepCycle. Nevertheless, in the revised manuscript we explicitly stated the potential utility of VAEs, as opposed to the AE implemented in DeepCycle, to model the distribution of spliced and unspliced reads:

“... . A similar approach can be implemented with a Variational Autoencoder (VAE). VAEs have already been applied as imputation methods to correct for capture rate and noise in scRNA-seq data^{21,29–31}. Though DeepCycle is not an imputation method, and is designed to detect 1-dimensional circular manifolds, VAE could allow the inference of a transcriptional phase together with the whole distribution of unspliced-spliced RNAs. ...”

Regarding the implementation of the VAE, the parameters of the neural network were optimized using the ELBO as an objective function as mentioned by the reviewer. As opposed to standard VAEs, we chose a von Mises distribution on the latent space to model cyclic boundary conditions. The attempts to train the network were inconclusive as it was not able to correctly fit the circular patterns in the unspliced-spliced RNA space. We do not exclude that future attempts may be more successful and will be included in a

newer version of DeepCycle. However, for the purpose of this paper, sorting cells according to their cell cycle state, we found the performance of a simple AE suitable and robust.

To improve the clarity and avoid any misunderstanding we have rewrote the text regarding the VAEs and moved it to the discussion section. We have also removed the mention of our attempts to fit a VAE. The new text in the discussion section is:

“Furthermore, we envision extending DeepCycle as a Variational Autoencoder (VAE), a neural network capable of modelling distributions over the input data. VAEs have already been applied on scRNA-seq data as imputation methods to correct for capture rate and noise^{21,29–31}. In our case, it will allow us to learn the posterior distribution of the transcriptional phase and model the whole distribution of unspliced-spliced RNAs.”

3. Authors mentioned that the lack of gold-standard labels for cell-cycle does not allow them to do proper method comparison. But in Figure 3A,C and Fig 4A they did label the cells with cell cycles based on marker genes. What I was asking in original comment was to define cells based on the marker genes and then the Theta to predict these cells.

In this new revision of the manuscript we have compared DeepCycle with two other methods that sort cells according to their cell cycle progression: Cyclone (Scialdone 2015) and Revelio (Schwabe 2002). Both methods rely on a predefined set of gene markers of the cell cycle phases. Cyclone assigns cells to broad cell cycle phases while Revelio sorts cells also in a continuous cell-cycle time similar in spirit as DeepCycle. We believe the comparison between the three methods highlights well the difficulty to address the reviewer’s comment. Depending on the markers/tools the assignment to the phases are different. For instance, Cyclone’s and Revelio’s phase assignments are not consistent. Furthermore, Cyclone’s assignments for the mESC (bottom left) are completely inconsistent with the DeepCycle and Revelio. Adjusted Rand Index (ARI) between DeepCycle and Cyclone assignments are -0.026 for mESC, 0.65 for ductal cell progenitors, and 0.25 for human fibroblasts. Finally, we cannot classify the cell phases from Revelio’s markers because these markers are assigned to mixed phases, e.g. G1/S, M/G1. However, overall, Revelio’s assignments are consistent with our phase subdivisions in all the datasets (see Supplementary Figure S9 attached below).

In DeepCycle, we used cyclin-E(1-2) peaks to define the G1/S transition (Alberts et al. MBC) and we exploit the decrease in total RNA counts to determine the beginning of mitosis. We used markers from Tyrosh et al. 2016 to set the S/G2 transition that is otherwise difficult to define based on few genes at the RNA level. This allows us to highlight transitions between broad cell-cycle phases. However, defining a standard list of marker genes for the different phases is out of the scope of our method. We believe that the major contribution of DeepCycle is instead to correctly sorting cells across the cycle revealing the continuous gene expression dynamics during the cell cycle. Nevertheless, our analysis suggests that there is no consensus about phase specific markers at the RNA level, consistently with the view that markers defined on different cellular models may vary, lacking in generalizability. We added this point to the Discussion:

“... . Given the variability in the cell cycle signatures among cellular models, defining the cell cycle phases in RNA data based solely on gene markers lacks generalizability. In the future, the usage of gene markers needs to be replaced by adopting methods relying on

dynamical features of gene expression, able to accommodate changes in the regulation of the cell cycle. ...”

On the other hand, the sorted cells across the cell cycle from DeepCycle and Revelio are in almost perfect agreement (below or Supplementary Figure S9), even if they are based on completely different methods. Notice though that Revelio still relies on marker genes for each of the transitions and phases to infer the cell cycle time. DeepCycle on the contrary, by exploiting information on unspliced and spliced RNA levels contained in the sequencing data, is able to select automatically for each cell-type (or condition) the subset of genes that have a cycling pattern in the unspliced-spliced RNA space.

Finally, we also tested other approaches. reCAT should also provide a cell cycle time, but it does not run on our datasets because of their large sizes (we contacted the authors to have further hints without receiving any answer). Oscope is not designed to return sorted cells but only clusters of cycling genes, therefore we were not able to make any comparison with it.

4. On line 463,464, the symbols do not show up properly on the pdf file probably due to conversion issue.

We apologize for the inconvenience and will make sure that the pdf visualizes the thetas properly.

Reviewer #2 (Remarks to the Author):

The authors have done a good job addressing the comments of the reviewers. The algorithmic details have sufficient depth in writing and in the figures. The github page and abstract are heavily extended / modified. Overall I think this work could be published in its current state, with minor fixes of spelling and grammatical mistakes. I have no further questions or concerns.

Stefan Bonn

We thank the reviewer for acknowledging our efforts to improve our manuscript. We appreciate very much his support for publication.

Reviewer #3 (Remarks to the Author):

All my concerns have been addressed. Besides, the authors also did great amount of additional work to significantly improve the quality of the paper, which I really appreciate. I thus would recommend the publication of the paper.

We are very happy to hear that the additional work has significantly improved the manuscript. We very much appreciate the reviewer's support for publication.

Two minor comments:

1. It seems that the theta cannot be displayed, e.g. line 123.

We apologize for the inconvenience and will make sure that the pdf visualizes the thetas properly.

2. The genes listed at the end of the main text file should be Supplementary Table S1 "cycling genes not yet considered in the GO term:cell_cycle that could be added as markers of the cell cycle", right?

It contains both the list of genes used to infer the phase in mESC and the one not yet annotated in the GO term:cell_cycle. We updated the table and added labels to each column to clarify their content.

Reviewer #5 (Remarks to the Author):

Review of Ms entitled "Cell cycle gene regulation dynamics revealed by RNA velocity and deep-learning" by Riba et al submitted to Nature Communications

Summary:

In this study, the authors use a computational analysis of deep scRNAseq data to assign the cell cycle phase of individual cells in unperturbed populations. Using RNA velocity, and parameters derived from the ratio of spliced to un-spliced transcripts they have projected the dynamic temporal activity of individual loci in single cells. The main finding is that the temporal oscillations of cell cycle-regulated genes can be detected by projecting a trajectory from (static) single time-point data of a cycling population to yield a specific signature of cyclic transcriptional activity in the "RNA velocity space". This index can be used to infer the cell cycle phase of individual cells. The study is performed in mouse embryonic stem cells (mESC) and somatic human fibroblasts (IMR90), and then tested using published data from ductal progenitor cells.

The study provides a potentially very useful tool for deriving cell cycle status from single cell transcriptomic data. Overall, the results are intriguing and the authors make some compelling arguments for the utility of this approach. However, there are a number of issues that require clarification, and potentially would make the study more accessible and therefore more widely applicable.

I will restrict my comments to the biology of the systems used, and the biological inferences drawn from the computational approaches. Overall, I found that the paper is written more for the computational biologist than the cell biologist. From the viewpoint of the biologists who might find the approach very useful, and in the interest of wider applicability, there are several statements/inferences/conclusions that would benefit from a more detailed explanation.

Example 1: The statement “The fits naturally generate gene expression series that can be analyzed to obtain detailed kinetic parameters”

An explicit statement is essential to explain how dynamic patterns can be computed from static single time-point data to infer cyclic activity, else it is difficult to appreciate the results.

Example 2: There is a very basic explanation of what TFs do, which I presume is for the computational expert, but the equivalent basic treatment of very complex computational inferences for the cell biologist are completely missing.

The Introduction does not provide the uninitiated with sufficient insight as to what ‘RNA space’ may refer to, why un-spliced vs. spliced reads are more useful than total reads, and how RNA velocity theory has come to be an accepted mode of investigation of transcriptomic data. Some explanation is found in the Discussion but in my opinion, this is better positioned in the Introduction so as to provide a clear framework from which to view this study. Can be reiterated in the Discussion if space is not an issue.

We thank the reviewer for pointing this out. We would like our paper to be accessible to the widest possible readership and especially to experimental biologists that may find our method useful to study gene regulation during the cell cycle. Therefore, we have significantly changed the introduction and include a detailed explanation of the idea behind RNA velocity to introduce the reader to the concept. We hope that the new text together with Fig 1F for some examples of RNA velocity patterns and Fig 2A for a scheme of the idea improves the understanding of our method.

From a biological perspective, it is not clear that the choice of three very different cell types is relevant. It would have been better to use comparisons between different states of a single cell type (for example undifferentiated mESC with their differentiated derivatives) as robust protocols exist for different lineages such as cardiac or neural or endothelial. From such data if the cell cycle changes that accompany differentiation (and cell cycle) could be mapped using DeepCycle, it would have been more convincing, and if then applied to mouse or human somatic cells such as fibroblasts or other cell types, one could appreciate the broader generalizability of the method.

However, given that substantial work has been done with the 3 very diverse cell types compared in the current ms., it would be important to tone down the claims of biological relevance and generalizability and restrict them to the specific results obtained.

We fully agree with the reviewer that there are very interesting and relevant questions that can be addressed by comparing the gene regulation dynamics during the cell cycle in mESCs and their differentiated derivatives. However, the main scope of this paper was to introduce a new computational method able to extract gene expression dynamics from scRNA-seq experiments and show its robustness and broad applicability. Thus, we believe that testing the performance of DeepCycle in three very different systems shows that the method is not tightly tailored to work on a particular cell type but on the contrary is flexible enough to adapt to different cellular systems. Remarkably, we obtain main differences in the cell cycle structure (different G1 lengths) without including any prior knowledge in the model showing the potential of DeepCycle to extract interesting

information regarding the cell cycle. More importantly, we are able to estimate gene expression continuous dynamics throughout the cell cycle genome-wide without the need of synchronizing or genetically modifying cells. This in principle opens the possibility to study gene regulation during the cell cycle in vivo, from model organisms to even patients. Finally, we predict transcription factors that may play an important role in regulating gene expression during the cell cycle which can be selected for further experimental validation. For all this we think the paper, although mainly computational, has certain biological relevance.

In this new revised manuscript, we checked carefully not to overstate the biological findings of our study. Whenever possible we stated clearly that our findings may suggest interesting biology but further experiments would be required to fully validate the results, for instance regarding the TF activity inference.

A major concern is the derivation of inferences about the behavior of G0 (quiescent) cells from a data set derived from a continuously cycling population.

The authors make a strong point about the application of their method to cells that have not been perturbed (synchronized) with drugs or by engineering with fluorescent markers. However, designation of a sub-population as G0/G1 based on FACS refers to the 2n DNA content (even a DNA/RNA ratio would have been more helpful to grossly distinguish G0 from G1). Data from cycling cells does not take into account the differences in behavior of truly quiescent cells (G0) which have entered into a qualitatively different state based on activation of a specific quiescence program coincident with cell cycle withdrawal. In this study, it is not at all clear that individual cells have actually reached a stationary phase, so it is a stretch to say whether cells are quiescent or will start cycling again, since no phenotyping in terms of kinetics of cell cycle have been done. Designation of a quiescent state relies on not just the DNA/RNA content or transcriptional profile, but critically, the kinetics and expression profile of cell cycle re-entry (the G0-G1 transition) which distinguishes these cells from G1 cells (even pre-R point), for discussions see Collier 2006; Goodell, 2004; 2006; Dhawan and Laxman 2015).

Therefore, the rather strong statements about whether cells are entering G0 are not justified, and need to be toned down or qualified substantially. G0 markers were derived from the Cheung and Rando data set, but these are from quiescent muscle stem cells (or very early after activation), not a continuously cycling population. It is not clear why the data from human fibroblasts was not used (Collier 2006) or from the core quiescence signature (Qsig) as derived by the Goodell group.

We thank the reviewer for pointing out these sets of markers of quiescence. We included the markers from Collier 2006 in the Supplementary materials (see figures below). They are consistent with the idea of an early G0-like state, and we extended the discussions about the nature of the non proliferative subpopulation. Our suggestion given the new evidence is that the cells are starting their transition towards quiescence.

Supplementary Figure S19

Supplementary Figure S21

G0 upregulated (Coller 2006)

Supplementary Figure S21(bis)

It is important to note that unperturbed populations of quiescent and exponentially cycling cells of the same type can be derived from normal and regenerating adult muscle (see Machado et al, 2017; van Velthoven et al, 2017), and might provide a much more stringent test of the ability of this computational approach to identify cells in different cell cycle phases, particularly G0.

We apologize for having misled the reviewer. Our method does not detect G0 directly, indeed we performed the cell cycle analysis on the proliferative subpopulation of fibroblasts. Once the cell cycle model is learned by DeepCycle, it can be run on the non proliferative subpopulation, to predict the closest cycling cells. This analysis shows that the non proliferative cells are transcriptionally more similar to cycling cells in mid-G1, which led us to investigate whether the non proliferative cells are related to quiescent cells. To avoid misleading future readers, the color of the non proliferative subpopulation in Fig. 5A has been changed to gray and the related text has been clarified.

From this perspective, several statements relating to Figure 5 therefore did not seem warranted or were overly emphatic:

Specifically:

(i) FoxM1 being designated as a marker of quiescence when it is specifically down-regulated after M phase is rather strange; normally one would designate an up-regulated gene to be a bona fide marker of a state if used alone; if used in conjunction with a signature, then up and down regulated genes provide more power. Further, since FoxM1 targets do not follow its expression, there may be many layers of regulation, which are not captured by this approach so it is not clear how this statement supports the conclusions.

FOXM1 is a marker gene listed in Collier 2006 and we added the corresponding reference in the text. The anticorrelation between FOXM1 expression and its targets may suggest its role as a transcriptional repressor. Indeed, experimental evidence indicates that FOXM1 may have different regulatory roles depending on the expressed isoform <https://doi.org/10.1515/BC.2007.159>. To further investigate and characterize the exact role of FOXM1 is out of the scope of our paper.

(ii) One would expect quiescence markers to be highly changed only in truly quiescent cells, not necessarily in the lagging cells of a G1 subpopulation of a cycling population, as their induction comes from extended period of time in G0, activating a new program. So this comparison did not seem compelling.

We extensively check markers of quiescence from Collier 2006 and Cheung and Rando 2013 and both show most of the genes consistent with our conclusion of G0-transitioning cells, we acknowledged in the text also the possibility of these cells representing other fibroblast states (Rognoni 2018):

“... . Therefore, nonproliferative fibroblasts might represent a differentiated state of the fibroblasts and not simply reflect cells entering in G0⁶⁹. ...”

(iii) While there is a reasonable case to be made for using DeepCycle, many new insights about the quiescence decision point have gained from experiments in the Meyer and Spencer labs, using unperturbed cycling populations of fibroblasts (albeit engineered with fluorescent markers, which I do not believe alters their conclusions). It would be important to acknowledge the utility of that approach.

We acknowledge papers linked to the cell cycle from the two labs in the discussion. Further, in agreement with one of their main findings, CDK2 level decreases in cells directed towards the quiescence-like state and increases in the cell undergoing a new cycle (<https://pubmed.ncbi.nlm.nih.gov/24075009/>). The figure below has been included as Supplementary Figure S23 and discussed in the section ‘Characterization of cycling cells shifting to a cycle-arrested state’.

CDK2 bifurcation

(iv) Inferring master regulators of G0 from this data feels like quite a stretch- as mentioned above, one cannot really derive strong conclusions about G0 since this is based on G1 cells from a cycling population.

After considering in the revised version the set of G0 markers in (Coller 2006) and (Cheung and Rando 2013) we obtained additional evidence suggesting that indeed we have a subpopulation of fibroblast cells that are transitioning towards a quiescent state. However, as already mentioned in the comment (ii), we stated in the manuscript that we cannot exclude alternative states of the fibroblasts (Rognoni 2018).

Therefore, the authors may want to substantially tone down their description and inferences from Fig5, and take into account the altered state of G0 cells compared to G1. Indeed, it is important to describe more carefully the entry into quiescence, its maintenance and exit from this type of data. But while some elements of the transition from G1 towards G0 may be captured from these data, I am not convinced that the G0 state is captured.

Since we do not think we captured fully quiescent cells, in agreement with the reviewer, and based on the latest analysis, G0 has been restated as an early G0 phase or cells transitioning to a quiescent state throughout the manuscript.

Can the authors do experiments to show that cells isolated on the basis of the RNA velocity index are kinetically distinct? Where do they fall in the cell cycle?

We are sorry but we are not sure to understand what the reviewer means by sorting according to the RNA velocity index. If the reviewer meant to sort cells according to different cell cycle phases, in the previous revision, we performed 3 bulk RNA-seqs on FACS-sorted mESCs in the three main phases G0/G1, S, and G2/M. The results are consistent with our sorting see Supplementary Figure S13. This reensure us to believe that our transcriptional phase estimated computationally is biologically meaningful. To be able to sort cells in a more fine-grained manner would require a different experimental approach which goes beyond the scope of the current paper.

Some specific issues include:

Is it not surprising that despite differences in read depth and very different transcriptional landscapes, all the data sets show ~15% unspliced reads? Would the ratio not be significantly different at different times/cells? Authors should comment on why regulation at splicing does not appear to be a feature, or provide a reference.

The ratio of unspliced reads is rather constant across the cell cycle with a small but detectable decrease during mitosis early G1 (see figure below). The signal is very noisy and it is hard to draw further conclusions.

However, the global pattern of splicing across the cell cycle is still unknown, see Petasny et al. 2021:

“... . The exact splicing patterns in each of the cell-cycle phases remain to be rigorously characterized. ...” in ref. PMID: 32950269.

Our analysis suggests that there is little change in global splicing compared to global transcription and degradation across the whole transcriptome and across the cell cycle. Interestingly, we observed a little decrease in the ratio of unspliced during mitosis, which could be related to the global downregulation of transcription occurring in this phase. However, due to the noise in the data it is difficult to carefully quantify this effect.

Fig1 # of genes per cell- is it not more correct to express as # of transcriptionally active genes per cell since the gene number per se should be identical for the two mouse cell types and slightly different for human cells?

We relabeled it as “Number of detected genes per cell” because of space constraints.

Fig3D does not exist in the revised version, but has been cited: harmonize

Thanks for pointing that out, we redirected the reader to the correct figure (Fig. 3C).

YY1 is both an activator and a repressor-how will that affect inferences?

In the case of TF which is both a repressor and an activator, the activity measures the average effect on its targets, so if most of the targets go up its activity will be positive otherwise, the opposite. We rephrased the corresponding sentence in the main text to reiterate that the TF activity is inferred from the change in expression of the targets: “..., Yy1/YY1 targets are upregulated in the G1 phase ...”

In conclusion, the ISMARA approach underestimates the role of TFs with a bivalent function. A more complex model would be required to correctly describe the action of these TFs. However, for TFs that behave mostly as activators or as repressors our

approach should be able to infer their activities accurately. In addition, it has been shown that the simplicity of the ISMARA approach avoids overfitting and yields robust results PMID: 24515121.

Fig5 why not use DNA-RNA ratio, which is a better separator of G1 from G0?

Thanks to the reviewer for pointing out this possibility we were not aware of. We checked the amount of RNA (UMI counts) in the non proliferative clusters and is comparable to the cycling cells (see figure below, UMI counts for proliferative cells in blue and non proliferative in orange). So we think that performing the flow cytometry experiments will return a very similar result. Considering this together with the G0 markers hinting at G0-transitioning cells, we speculate that these cells did not have the time yet to decrease their RNA production and did not yet reach a fully quiescent state.

Statement: Ybx1 expression is linked to poor prognosis in pancreatic ductal adenocarcinoma –why is this relevant in the context of this paper?

This information was added to give context to the role of Ybx1 in ductal cell progenitors and has been removed in the last version of the manuscript. Ybx1 regulates cell cycle progression and proliferation, because of that it is typically associated with papers analyzing its function in pancreatic cancers (e.g. PMID: 32300640).

Line 308 "G1 are bigger than the cells in S and G2/M suggesting that the cells waiting in G0/G1 are increasing in size, it remains unclear whether they will re-enter the cycle later. The cell velocities are consistent with our interpretation and do not clarify if the cells in G0 will start cycling again" (Supplementary Figure S16).

Again: designating G1 cells from a cycling population as quiescent is not warranted as they just represent a continuum of lagging G1 not G0

See previous answers

Need explicit statement of why spliced to unspliced transcripts is a better measure of transcriptional status of a given locus than total transcripts

Notice that spliced mRNA levels result from the balance between synthesis and degradation. Thus, mRNA stability can obscure changes in the transcriptional status of a

gene. Unspliced mRNA, being a measure of pre-mRNAs, are more directly linked to active transcription assuming a fast splicing process compared to the stability of the spliced mRNA. Indeed, if the stability of spliced mRNA is larger than the splicing time, the unspliced reads will accumulate faster than spliced reads when the gene is activated. On the contrary, when the gene is deactivated the unspliced reads will decrease faster. For cell-cycle regulated genes, we expect to observe an activation and deactivation phase leading to a cycling pattern in the unspliced-spliced RNA space. Therefore, comparing spliced and unspliced reads in each single cell allows to disentangle the contribution of synthesis and degradation in gene expression. We have included a paragraph in the Introduction to clarify these ideas.

Comparison of ESC and somatic cells is not strictly useful since they have a very different cell cycle structure and regulation and mESC never enter G0

As mentioned above, our main goal of the comparison was to show the approach is reliable on different cellular models and to ensure its broad applicability. Further, studying the cell cycle in very different cell lines facilitates the validation process, highlighting macroscopic differences in the cycles, e.g. phase lengths, and specific genes related to a specific model, e.g. Ybx1 in the ductal cells.

Clarifications required for the following:

Abstract: rather than pluripotent vs. differentiated cells more correct to say embryonic vs. somatic since from the cell cycle perspective both are cycling but with different characteristics

We modified the text as suggested by the reviewer.

Supp Fig 1: indicate what the blue and orange traces in the gene expression plot (lower panel) represent. The current figure legend has no explanation. This figure is skewed for genes that are highly expressed in cycling cells, where only Dek represents a gene more highly expressed in non-cycling cells- more examples of these should be shown; genes that are known to be expressed only in S phase or early G1 could also be shown to give a sense of the robustness of this approach.

Supplementary Figure 1 includes the most differentially expressed genes between the two subpopulations. We added a new Supplementary Figure (S2) with some markers of the G2/M, S (Schwabe 2020) and G0 phases (Coller 2006, Cheung and Rando 2013). Figure reported below.

Supp figs often refer to main figs (eg S17), but they should be self-explanatory, and the legends should be more detailed to allow appreciation of the point being made. Perhaps a title based on the inference being drawn from each sup fig can be included to enable understanding?

Specific titles summarizing the supplementary figures have been added.

Reviewers' Comments:

Reviewer #5:

Remarks to the Author:

The revised manuscript has addressed all the issues of concern, and has provided substantial clarity to the design of experiments and conceptual framework. The additional supplementary figures also help to locate the work in context.

I can recommend publication.